# Satellite-derived global ocean phytoplankton phenology indices.

Sarah-Anne Nicholson[1], Thomas J. Ryan-Keogh[1], Sandy J. Thomalla[1,2], Nicolette Chang[1,5], Marié E. Smith[3,4]

[1]Southern Ocean Carbon-Climate Observatory, CSIR, Cape Town, South Africa

[2]Marine and Antarctic Research Centre for Innovation and Sustainability, University of Cape Town, Cape Town, South Africa

[3]Coastal Systems and Earth Observation Research Group, CSIR, Cape Town, South Africa

[4]Department of Oceanography, University of Cape Town, Cape Town, South Africa

[5]Global Change Institute, University of the Witwatersrand, Johannesburg, South Africa

*Correspondence to*: Sarah-Anne Nicholson (snicholson@csir.co.za)

## Abstract

Phytoplankton bloom phenology is an important indicator for the monitoring and management of marine resources and the assessment of climate change impacts on ocean ecosystems. Despite its relevance, there is no long-term and sustained observational phytoplankton phenological product available for global ocean implementation. This need is addressed here by providing a phenological data product (including among other seasonal metrics, the bloom initiation, termination, duration, and amplitude timing) using satellite-derived chlorophyll-a data from the Ocean Colour Climate Change Initiative. This multi-decadal data product provides the phenology output from three widely used bloom detection methods at three different spatial resolutions (4, 9 and 25 km) allowing for both regional and global-scale applications. When compared to each other on global scales, there is general agreement between the detection methods and between the different resolutions. Regional differences are evident in coastal domains (particularly for different resolutions) and in regions with strong physical-biogeochemical transitions (notably for different detection methods). This product can be used towards the development of national and global biodiversity assessments, pelagic ecosystem mapping and for monitoring change in climate sensitive regions relevant for ecosystem services. The dataset is published in the Zenodo repository under the following DOIs, 4 km: https://doi.org/10.5281/zenodo.8402932, 9 km: https://doi.org/10.5281/zenodo.8402847 and 25 km: https://doi.org/10.5281/zenodo.8402823 (Nicholson et al., 2023a, b, c) and will be updated on annual basis.

## 1 Introduction

The seasonal proliferation of phytoplankton across the world's ocean is a ubiquitous signal visible from space, and one that plays a crucial role in the Earth system. Phytoplankton "blooms" capture 30-50 billion metric tons of carbon annually, representing almost half of the total carbon uptake by all plant matter (Buitenhuis et al., 2013; Carr et al., 2006; Falkowski, 1994; Field et al., 1998; Longhurst et al., 1995). Their key role in driving the strength and efficiency of the biological carbon pump, the transfer of atmospheric carbon to the deep ocean interior, is a crucial component of the global carbon cycle and instrumental in the assessment of climate feedbacks and change (DeVries, 2022; Henson et al., 2011). Phytoplankton also mediate climate through the production of important atmospheric trace gases such as nitrous oxide, a potent greenhouse gas, and volatile organic carbons such as dimethyl sulphide, that have a significant impact on cloud formation and global albedo (Charlson et al., 1987; Korhonen et al., 2008; McCoy et al., 2015; Park et al., 2021). As the foundation of the marine food chain, phytoplankton are critical to supporting higher trophic levels and a lucrative fisheries industry that impacts global food security (Gittings et al., 2021; Stock et al., 2017). There is an enormous benefit to society in being able to predict ecosystem responses to environmental change, by providing the knowledge necessary for competent decision-making. As such understanding, characterising and accurately predicting changes in the annual cycle of phytoplankton blooms provides an essential tool for managing marine resources and for predicting future climate change impacts (Thomalla et al., 2023; Tweddle et al., 2018).

Phytoplankton phenology refers to the timing of seasonal activities of phytoplankton biomass and is used widely as an indicator to characterise phytoplankton blooms and to monitor their variability over time. Adjustments in the characteristics of phenology typically reflect alterations in ecosystem function that may be linked to environmental pressures such as climate change (Henson et al., 2018; Racault et al., 2012; Thomalla et al., 2023). Key phenological phases of phytoplankton bloom development include: the time of initiation, the time of maximum concentration (amplitude), the time of termination and duration as the time between initiation and termination. These phytoplankton bloom phases are typically driven by seasonal changes in physical forcing (such as incoming solar radiation, water column mixing and nutrient depletion), which are generally linked to large-scale climate drivers (Racault et al., 2012; Thomalla et al., 2023). The timing of the bloom initiation and amplitude is particularly critical for efficient trophic energy transfer, which can be impacted negatively through trophic decoupling. For example, mismatches between bloom timing and zooplankton grazing can lead to suboptimal food conditions for higher trophic levels which in turn has been linked to the collapse of crucial fisheries (Cushing, 1990; Koeller et al., 2009; Seyboth et al., 2016; Stock et al., 2017). Bloom duration impacts the amount of biomass being generated within a season that can be exported to the ocean's interior or transferred to higher trophic levels via the marine food web and can thus play a more important role than bloom magnitude (Barnes, 2018; Rogers et al., 2019). Bloom timing has also been shown to influence the seasonal cycles of $CO_2$ uptake, primary production and the efficiency of carbon export and storage (Bennington et al., 2009; Boot et al., 2023; Lutz et al., 2007; Palevsky and Quay, 2017). Having access to a global data product that characterises the seasonal cycle of phytoplankton over the last 25 years and into the future can thus provide a valuable tool to users that require an understanding of key aspects of the growing season and how these may be changing over time.

Current generation Earth System Models (ESMs) show that phytoplankton phenology is changing and will continue to change in response to a warming and more stratified ocean (Henson et al., 2018; Yamaguchi et al., 2022). For example, blooms are predicted to initiate later in the mid-latitudes and earlier at high and low latitudes by ~5 days per decade by the end of the century (Henson et al., 2018). But what about changes in bloom phenology in the contemporary period? Satellite-based ocean colour remote sensing, which provides estimates of chlorophyll-a (chl-a) concentrations (a proxy for phytoplankton biomass), is the only observational capability that can provide synoptic views of upper ocean phytoplankton characteristics at high spatial and temporal resolution (~1 km, ~daily) and high temporal extent (global scales, for years to decades). In many cases, these are the only systematic observations available for chronically under-sampled marine systems such as the polar oceans. In 1997, the first global ocean colour observing satellite (SeaWiFS) was launched and these observations have been sustained through a successive series of additional ocean colour satellites (MODIS, MERIS, VIIRS, OLCI). These have all been merged by the European Space Agency (ESA) into the Ocean Colour Climate Change Initiative (OC-CCI) satellite-derived data product, which provides ~26 years of ocean colour data, with substantially reduced inter-sensor biases, for climate change assessment (Sathyendranath et al., 2019). We note however that despite their obvious spatial and temporal advantages, remotely detected water-leaving radiances emanate from only the first optical depth and give little quantitative information about the vertical structure of the water column, which can be particularly important in low nutrient regions where a subsurface chl-a maxima is prevalent (Stoer and Fennel, 2024). In addition, we recognise that the OC-CCI chl-a satellite-derived data product may exhibit regional biases (that can vary in both magnitude and direction) and arise from several factors inherent to both satellite remote sensing technology and the complexities of ocean ecosystems. One example is that algorithms are often regionally trained on datasets from specific parts of the world, which can result in discrepancies when applied globally. Despite these regional biases, satellite ocean colour chl-a observational data products remain highly valuable, especially when the goal is to identify patterns in the seasonal cycle of phytoplankton and how these patterns evolve over time. While local accuracy may be impacted by biases, the broader trends—such as the timing of spring blooms, the intensity of summer productivity, or the length of growing season—are still well captured. This is because biases tend to be relatively consistent over time in any given region, allowing researchers to focus on changes in these patterns rather than on the absolute values. These long-term changes in the seasonal cycle are crucial for understanding how marine ecosystems respond to environmental stressors like warming temperatures, ocean acidification, and changes in nutrient availability.

The estimation of phytoplankton phenology from OC-CCI remote sensing of chl-a can provide important information of the rates of change in key indices on a global scale for comparison to those derived from ESM's. For example, a recent study by Thomalla et al., (2023) determined the trends in phenology metrics in the Southern Ocean using 25 years of satellite-derived chl-a (1997-2022) data. Their results revealed that large regions of the Southern Ocean expressed significant trends in phenological indices that were typically much larger (e.g. <50 days decade$^{-1}$) than those reported in previous climate modelling studies (< 5-10 days decade$^{-1}$), which suggests that ESM's may be underestimating ongoing environmental change. Thomalla et al., (2023) conclude, that seasonal adjustments of this magnitude at the base of the food web may impact the nutritional stress, reproductive success, and survival rates of larger marine species (e.g., seals, seabirds, and humpback whales), in particular if they are unable to synchronise their feeding and breeding patterns with that of their food supplies. It is anticipated

that a similar analysis using these key phytoplankton metrics applied to the global ocean or specific regions of interest will reveal regional sensitivities of ecosystems to change with important implications for ecosystem function and associated societal impacts. There is also a need for the continuous monitoring and ongoing assessment of the seasonal adjustments of phytoplankton on global scales (in addition to continued benchmarking for ESMs), which would require regular updates of key phenological metrics going forward. Such information is relevant for effective marine management programs and early detection of vulnerabilities in key regions, e.g., those necessary for sustaining fisheries. In addition, a phenology data product such as this can provide a useful aid for the planning of oceanographic research campaigns that wish to align with or determine their occupation relative to key aspects of the growing season. Finally, this data product could also be valuable to support those users without the programming know-how or access to computationally expensive resources that are required to generate it.

Here we present a new global phytoplankton phenological data product with indicators that include among other metrics bloom initiation, termination, amplitude and duration. These metrics are computed using three different gridded resolutions (4, 9 and 25 km) and with three different methodologies of determining phenology. This satellite-derived data product facilitates the global characterisation of the climatological seasonal cycle and can be used to identify the sensitivity of the seasonal cycle to change (through the analysis of trends and anomalies). The phenology data product is currently available from 1997 until 2022 and will be updated annually and in sync with any version updates of the OC-CCI chl-a data product.

## 2 Methodology

### 2.1 Data and pre-processing

Satellite-derived chl-a concentrations (mg m$^{-3}$) were obtained from the ESA, from OC-CCI (https://esa-oceancolour-cci.org; Sathyendranath et al., 2019) at 4 km and 8-day resolution. The latest available OC-CCI product (version v6.0, released on 04/11/2022) is used in this present study. This version marks a substantial change to previous versions (e.g., v5.0, see Sathyendranath et al., (2021)) in that it incorporates Sentinel 3B OLCI data, the MERIS-4$^{th}$ reprocessing dataset, upgraded Quasi-Analytical algorithm (QAAv6) and the exclusion of MODIS and VIIRS data after 2019 (refer to D4.2 - Product User Guide for v6.0 Dataset from https://climate.esa.int/en/projects/ocean-colour/key-documents/ for further details on processing and validation). The OC-CCI data product was generated with the specific aim of studying phytoplankton dynamics at seasonal to interannual scales. Indeed, it has been used widely by the scientific community for studying phytoplankton phenology (e.g., Anjaneyan et al., 2023; Delgado et al., 2023; Ferreira et al., 2021; Gittings et al., 2019, 2021; Racault et al., 2017; Silva et al., 2021; Thomalla et al., 2015, 2023). Data provided by OC-CCI covered the period from 29/08/1997 – 27/12/2022 for the global ocean (90°N – 90°S and 180°E – 180°W).

The phenological indices described below are calculated using three horizontal resolutions in surface chl-a, the native 4 km resolution as provided by OC-CCI and a regridded 9 km and 25 km horizontal resolution. The 4 km

and 9 km resolutions are considered important for smaller-scale regional needs such as coastal applications and field campaigns. The 25 km resolution is the most computationally efficient for users to work with, as it results in a reduction of missing data and is useful for global open-ocean applications. For the 9 km and 25 km resolutions, chl-a is regridded onto a regular grid through bilinear interpolation using the xESMF Python package (Zhuang et al., 2023). In all resolutions for phenological detection, data gaps were reduced further by applying a linear interpolation scheme in sequential steps of longitude, latitude, and time (Racault et al., 2014). A two-point limit (e.g., the maximum number of consecutive empty grid cells to fill) is chosen for the interpolation to avoid overfilling of regions that contain larger coherent data gaps. We further apply a 3 time-step (24 days) rolling mean along the time dimension to avoid any outliers that may result in fake detection points. However, for the Seasonal Cycle Reproducibility (SCR) computations only interpolation (time, lat and lon) is carried out, this is discussed further below.

## 2.2 Phenological Indices and Detection

Phytoplankton blooms typically manifest as a seasonal cycle, with a bloom initiation that identifies the timing of the ramp up in phytoplankton growth and biomass accumulation followed by bloom peaks within the growing season (which could be multiple) and finally the bloom termination, which defines the end of the growing season. The phenological indices applied here are based on those applied to the Southern Ocean in Thomalla et al., (2023). To calculate the phenological indices for initiation and termination, we apply three main detection methods used by the community (e.g. Brody et al., 2013; Ji et al., 2010), which are detailed below (iii and iv). Each detection method has its strengths and weaknesses, and therefore the choice of method for application can be determined by the user needs, which are elaborated on in Brody et al., (2013). These methods were chosen over other approaches (e.g. Platt et al., 2009; Rolinski et al., 2007) due to the method's suitability for estimates across global scales as it is capable of encompassing a wide range of different shapes in phytoplankton blooms (Racault et al., 2012). Below we outline the series of steps implemented for estimating the global phenological indices and provide an accompanying flow chart (Figure 1) to illustrate the succession of steps being implemented. In addition, we provide some example applications at four key observing stations (Figure A1) to facilitate a visualisation of the derived phenological indices from four annual time series.

(i) Bloom maximum climatology: The climatological peak (maximum amplitude) of the bloom was identified as the local maximum in chl-a occurring within each grid cell's 25-year climatology. This approach was necessary because the timing of bloom events varies globally, i.e., southern hemisphere blooms typically occur during austral spring - summer (September - February), while northern hemisphere blooms occur in boreal spring - summer (April - August) (Racault et al., 2012). Furthermore, both hemisphere tropics tend to be approximately 6 months out of phase with both hemisphere higher latitude regions. As such, it would be inappropriate to use a fixed date period (or "bloom slice" see below) to identify bloom occurrence on global scales. Instead, for each grid cell we calculate the 8-day mean climatology. The date of the maximum climatological bloom for each pixel is then used to centre the timing of the phenology detection methods described below.

(ii) Identification of bloom peaks: For every pixel on a year-by-year basis we take the climatological bloom maximum peak ±6 months and determine the date and magnitude of the bloom maximum peak for each year. To ensure that seasonal blooms with more than one peak could be accounted for, multiple bloom peaks were defined as a second, third, or $n^{th}$ local maxima where the chl-a concentration reached at least 75% of the amplitude of the bloom maximum peak magnitude and were a minimum of 24 days (i.e., 3 x 8 day time intervals) away from the bloom maximum peak for that year. The 75% threshold was chosen to identify peaks with similar magnitude to the bloom maximum peak so as to allow for the occurrence of a multiple peak growing season. Choosing a threshold higher than this would likely exclude recognisable bloom peaks (which could lead to an underestimate of the bloom duration), while choosing a lower threshold may include sub-seasonal variability and lead to an overestimation of the bloom duration. These additional peaks were found within ±6 months of the maximum peak. An example of such a multi-peak bloom detection is provided in Figure 1 and Figure A1c. The additional peaks were identified with the Python SciPy (Virtanen et al., 2020) function 'find_peaks'.

(iii) The 'bloom slice': The bloom slice, used to find the bloom initiation and termination dates, is identified for each pixel as the 6-month time span preceding and following from the maximum bloom peak (ii). Or in the case of multi-modal blooms, 6-months preceding the first and following the last peak respectively.

(iv) Bloom initiation: The bloom initiation date for each bloom slice as described in (iii) is calculated as the first date before either the bloom maximum, or the first peak in the event of multi-modal blooms, according to the following thresholds:

1. *Biomass-based threshold method (TS)*: First determine the range as the difference in chl-a concentration between the bloom maximum and preceding minimum. Then identify the bloom initiation as the first date that the chl-a concentration was greater than the minimum chl-a concentration plus 5% of the chl-a range.

2. *Cumulative biomass-based threshold method (CS):* First remove any values preceding the bloom slice minimum chl-a concentration and any values greater than 3 times the median of the bloom slice, before calculating the cumulative sum of chl-a. Then identify the first date that the chl-a concentration was greater than 15% of the total cumulative chl-a concentration.

3. *The rate of change method (RC):* First determine the rate of change of the bloom slice and then identify the first date after the minimum that the chl-a rate of change was greater than 15% of the median rate of change in chl-a concentration.

To note, the choice of above chosen percentage thresholds are in accordance with those used by previous phenological detection studies (Brody et al., 2013; Hopkins et al., 2015; Ji et al., 2010; Thomalla et al., 2011, 2015, 2023).

(v) Bloom termination: The bloom termination date for each bloom slice was similarly calculated as the first date after the bloom maximum, or the last peak in the event of multi-modal blooms, according to the following thresholds:

1. *TS*: the first date that the chl-a concentration was less than the minimum chl-a concentration plus 5% of the chl-a range.

2.   *CS:* the first date between term peak and post bloom minimum that the chl-a concentration was less than 15% of the total cumulative chl-a concentration.

3.   *RC:* the first date between term peak and post bloom minimum that the chl-a rate of change was less than 15% of the median rate of change in chl-a concentration.

(vi) Bloom duration: The bloom duration was calculated as the number of days between the bloom initiation and termination dates. This is applied to each phenological detection method described above (TS, CS and RC).

(vii) Integrated and mean bloom chl-a: The seasonally integrated bloom chl-a was calculated using the NumPy (Harris et al., 2020) trapezoidal function as the chl-a concentration integrated between the bloom initiation and termination dates. The seasonal mean chl-a was calculated as the average chl-a between the bloom initiation and termination dates. These are applied to each of the three phenological detection methods described above (TS, CS and RC).

(viii) SCR: The variance of the seasonal cycle was calculated as defined in Thomalla et al. (2023), where the SCR is the Pearson's correlation coefficient of the annual seasonal cycle correlated against the climatological mean seasonal cycle. A value of 100% is indicative of an annual seasonal cycle that is a perfect repetition of the climatological mean, while a value of 0% means that there is no annually reproducible mean seasonal cycle. Unlike for phenological indices i-vii, for SCR the original OC-CCI v6.0 data were used for the three different grid resolutions, however with only spatial-temporal interpolation for gap filling and no rolling mean to avoid smoothing out temporal variability. For SCR for each pixel the bloom slice is restricted to 12 months (i.e., January to December).

The cyclical nature of the year day calendar presents a significant challenge when calculating means and standard deviations of phenological indices. For example, we need to avoid a situation where the mean bloom initiation between a year with a bloom in December (day of year = 340) and a year with a bloom in January (day of year = 10) is incorrectly calculated as an average bloom initiation date in July (day of year = 175). To address this, as similarly applied in Thomalla et al. (2023), we used the Python SciPy function circmean (or circstd for standard deviation), which calculates circular means for samples within a specified range, correctly identifying the mean as day of year 357. The user should also be aware that any pixels in the first year of this satellite-derived data product where the initiation date is the same as the first available start date of chlorophyll-a (e.g. 04-09-1997) should be masked out. Similarly, any pixels in the last year of the product where termination date is the same as the last available chlorophyll-a time-step (e.g. 27-12-2022) should be masked out.

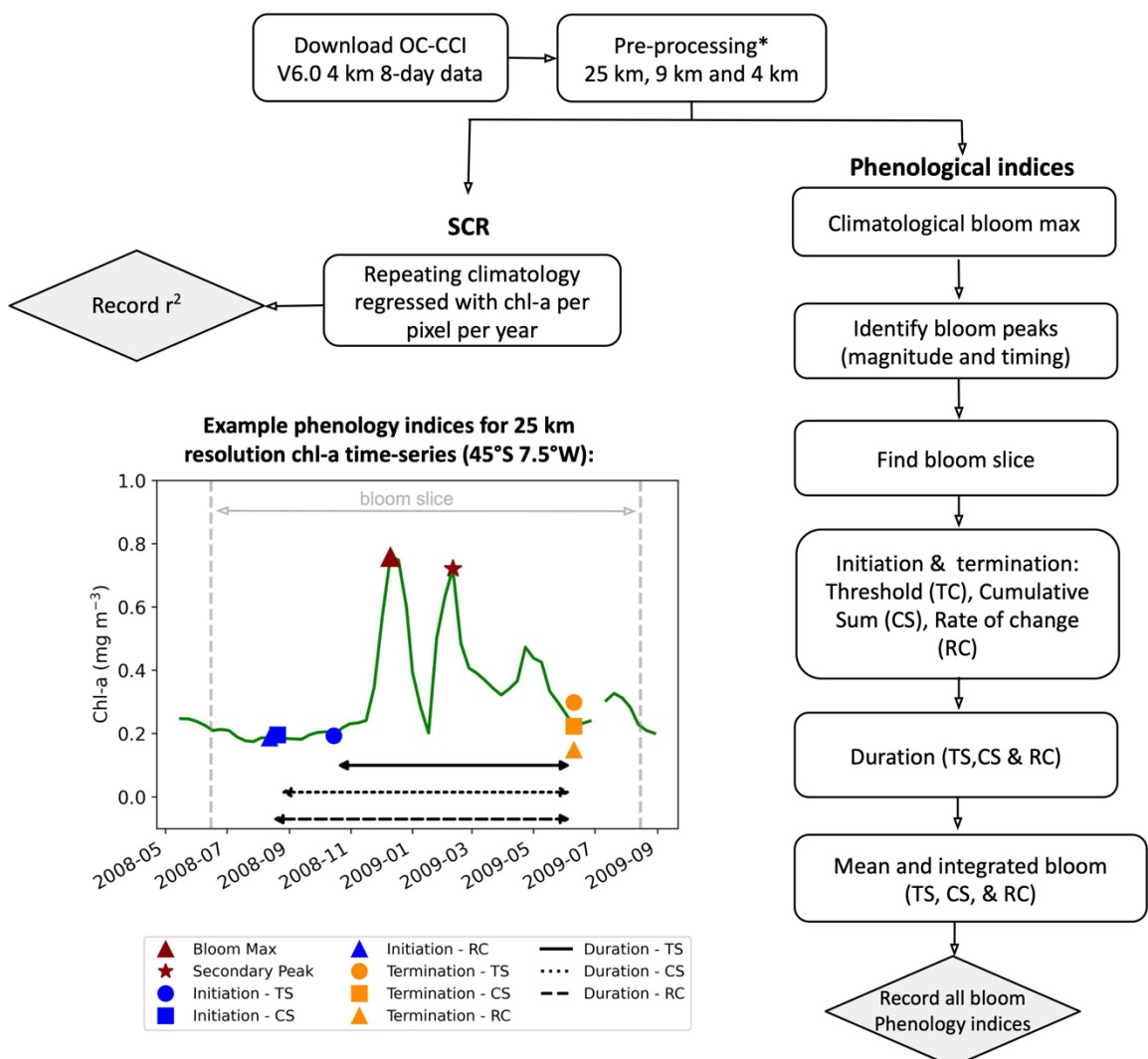

Figure 1: Methodological flow chart outlining the steps taken to calculate the phytoplankton seasonal metrics. An example time-series from ocean color satellite observations from OC-CCI illustrating the performance of the resulting phenological indices for a bimodal (double peak) bloom in the Southern Ocean (45˚S, 7.5˚W) is provided for the three different phenological methods, biomass-based threshold (TS), cumulative sum (CS) and rate of change (RC). *See Methodology for pre-processing steps.

**3 Results and Discussion**

**3.1 Global open-ocean phytoplankton seasonal metrics**

A significant degree of regional variability is evident in the mean distribution of seasonal metrics (bloom amplitude, timing, and seasonality) (Figure 2). Bloom magnitude metrics (max bloom chl-a, mean bloom chl-a and integrated bloom chl-a; Figure 2a-c) are all higher in the high-latitudes and in the coastal regions, particularly

in the Eastern Boundary Current Systems, and lowest in the oligotrophic subtropical gyres. There is a general equator-to-pole symmetry in the timing of phytoplankton blooms between the northern and southern hemispheres. In the subpolar regions phytoplankton blooms initiate in the northern hemisphere during Boreal Spring to early summer (March-May) and in the southern hemisphere in Austral Spring to early summer (September-November) in response to light availability (Sverdrup, 1953) (Figure 2d). While in the subtropics, where there is ample light throughout the year, blooms typically initiate in autumn to winter in response to nutrient supplies through winter-driven deepening of the mixed-layer (Fauchereau et al., 2011; Thomalla et al., 2011). In both the Antarctic and Arctic polar regions, phytoplankton blooms initiate in Austral (December) and Boreal summer (July), when the sea-ice cover melts. The timing of bloom maximum follows a similar equator-to-pole symmetry as bloom initiation (Figure 2g), with high latitude regions peaking in Austral and Boreal summer, whereas the subtropics peak in Austral and Boreal winter. This large-scale meridional structuring of the bloom timing is as expected and similarly found in previous large-scale satellite based phenological studies (Kahru et al., 2011; Racault et al., 2012; Sapiano et al., 2012). There is a larger degree of spatial heterogeneity in bloom termination (Figure 2e), particularly evident in regions such as the high latitude North Atlantic and sub-Antarctic, with terminations that extend up to 6 months later in comparison to surrounding areas which were initiated at a similar time. This manifests in zonal asymmetries across the different basins for bloom duration (Figure 2f), with considerably longer blooms occurring in the Pacific basin compared with the Atlantic and Indian basins. SCR covers a large range of variability across latitudinal bands. Notably, SCR (Figure 2h) is oftentimes low in regions where bloom duration is long, and this relationship is strongest in the tropical Pacific (r ~ -0.4). In these oligotrophic regions, where bloom amplitude is constrained by nutrients, the seasonality of phytoplankton blooms is not well-defined and characterised by high intraseasonal variability (Figure 2, Thomalla et al., (2011)). Worth noting when applying our bloom detection method to these regions is that it does not constrain a bloom slice to be within a 12-month period, as is done in other phenology studies (e.g. Henson et al., 2018). Rather, by allowing for multiple peaks to be considered within a bloom, this approach may produce extended bloom durations that are beyond a year in regions with no discernable or strongly defined seasonal cycle. In the Southern Ocean, with higher bloom amplitudes and a well-defined yet highly variable seasonal cycle, sustained blooms of ~250 days are detected, which have been attributed to intermittent physical forcing (high-frequency wind and meso to submesoscale dynamics) that entrain nutrients and prolong the seasonal termination (Thomalla et al., 2011, 2023).

A comparison of our satellite-derived phenology product with bloom indices derived from in situ data at a selection of regional case studies shows reasonable agreement. For example, in the Saronikos Gulf (Eastern Mediterranean), Kalloniati et al., (2023) report a mean bloom initiation in early October (2005–2015), which compares well with our mean bloom initiation over the same period of 24 September. Similarly, their mean bloom peak occurs in late February, closely matching our estimate of 24 of February. However, there are notable differences in bloom termination with their approach reporting a seasonal bloom that terminates in mid-April, compared to our estimate of ~100 days later on 13 July. This discrepancy likely arises because their method does not account for multiple bloom peaks, whereas our method is specifically designed to include the secondary peak observed in April as part of the seasonal bloom (see their Figure 3c). Another example from long-term mooring observations (1998-2022) in the Bering Sea shelf (Nielsen et al., 2024) reports the timing of the bloom maximum

to range annually between the end of April to mid-June (see their Figure 2), which compares well with our mean
estimate over the same period of 25 of May (standard deviation of 57 days). In a Red Sea comparison, although
our satellite derived phenology data product was able to detect similar bloom initiation and max peak timing for
the primary bloom in winter (as observed by Racault et al., 2015), it is not designed to provide indices fort bi-
modal blooms and thus is unable to identify the secondary bloom in summer. Beyond these existing studies, we
applied our phenological detection method (TS) to chlorophyll-a data from the Hawaii Ocean Time-series (HOT)
and Bermuda Atlantic Time-series Study (BATS) long-term monitoring sites (Figure A2, Valente et al., 2022). At
HOT (1998-2018)(Figure A2a), the in situ bloom initiation occurred on 25 July (±48 days) compared to the
satellite-derived occurring on the 21 July (±42 days), in situ bloom max timing on 12th of December vs. 5th of
December, and termination on 22 May (±32 days) vs. 6 June (±29 days) and duration in situ of 299 days vs
durations of 303 days from satellite data. Similar agreement was seen in the BATS station (Figure A2b).

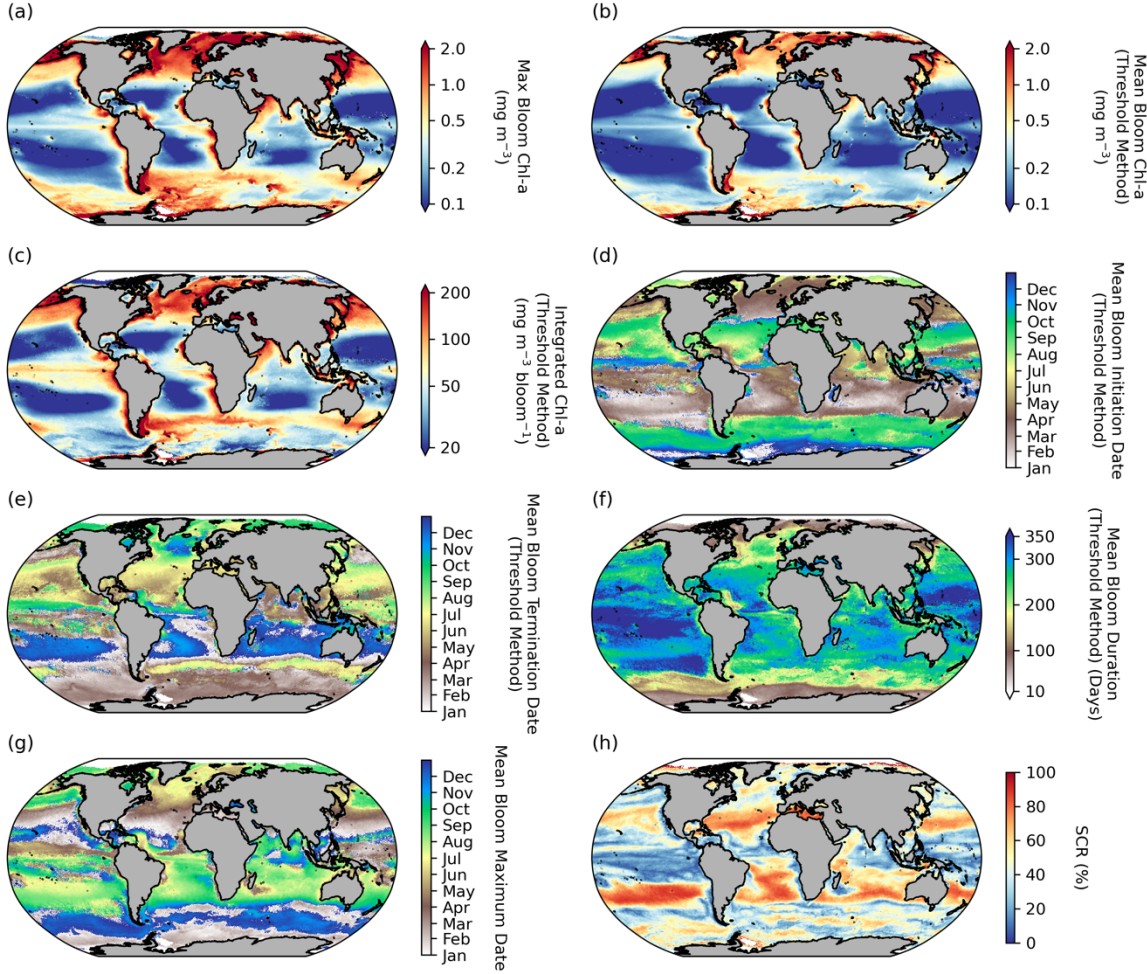

Figure 2: Global distribution of phytoplankton seasonal metrics. Mean [1998 – 2022] maps of (a) bloom max
chlorophyll (chl-a), (b) mean chl-a over bloom duration, (c) integrated chl-a over bloom duration, (d) bloom
initiation, (e) bloom termination, (f) bloom duration, (g) bloom max chl-a date, and (h) seasonal cycle
reproducibility (SCR). Phenological indices (b-f) are determined using the Biomass-based threshold method as
defined in Henson et al., (2018); Thomalla et al., (2023).

**3.2 Comparison between phenology detection methods**

Phytoplankton blooms can initiate rapidly, slowly, be short lived, intermittent, or sustained over a growing season, with different detection methods being more or less sensitive to these varying characteristics of the seasonal bloom (Brody et al., 2013; Ji et al., 2010; Thomalla et al., 2023). In this satellite-derived data product we have chosen to provide three widely used bloom detection methods for all three resolutions allowing the user to determine which method (or all) is most appropriate for their region and application (Figure 3 and Figure A3). Indeed, these methods each have their strengths and weaknesses. For example, as explained in Brody et al., (2013), the biomass based TS method will likely capture the bloom start dates at the largest increase in chlorophyll concentrations. It is thus more suitable for studies wanting to investigate the match or mismatch between phytoplankton and upper trophic levels as the match-mismatch hypothesis is based on the timing of the high phytoplankton biomass period (Cushing, 1990). This method has been found to be relatively insensitive to the percentage of the threshold used (Brody et al., 2013; Siegel et al., 2002). The RC method, which identifies the bloom initiation as the time when chl-a increases rapidly, is likely more suitable for investigating the physical or biochemical mechanisms that create conditions in which the bloom occurs (Brody et al., 2013). Whereas the CS method could be used to identify either of the features above, Brody et al. (2013) showed that, while there are sensitivities of the CS method to the threshold chosen, the 15% threshold as applied here, is most appropriate at capturing bloom initiation dates of both subpolar and subtropical regions and thus most appropriate to be applied across global scales. It is interesting and potentially valuable to determine when and where different methods of determination agree or disagree, and we advocate for users to apply all three methods so that they may interrogate the differences and make informed decisions about choosing one over another or utilising all three to define a range in the desired metric. In Figure 3, the standard deviation (STD) between the three methods is applied globally to assess the agreement between climatological means from the different methods.

Across large regions of the global ocean, there is good agreement between the different methodological approaches (e.g. the global mean STD for the phenological timing indices is ~8-days) (Figure 3 a- b) All methods produce similar large-scale patterns (Figure A3 a-c, g-f, m-o). There are however some specific regions where larger differences in timing emerge of ~30-50 days (Figure 3 and Figure A3 d-f, j-l), which are of a similar order of magnitude as reported by Brody et al., (2013) who found areas with differences exceeding two months. The largest differences for both bloom initiation and termination tend to coincide with transitional zones such as at the boundaries between the subtropical and subpolar gyres in both hemispheres and in all three basins (Figure 3a,b). This is not too surprising, given that these boundaries represent areas of significant biogeochemical signatures and regime shifts between phytoplankton seasonal characteristics with strong north-south gradients in bloom metrics (Figure 2). While there are no other comparisons of these detection methods on a global scale, such differences were similarly seen in Brody et al. (2013) for the North Atlantic bloom, their Figure 4, where the largest differences between bloom initiation methods occurred at the sharp transition boundaries between the subtropical and subpolar latitudes. In general, there is stronger agreement between methods in the higher subpolar latitudes compared to subtropical latitudes, as evidenced by slightly elevated STDs in the subtropical gyres (Figure 3a,b). The subtropical oligotrophic regions are characterised by phytoplankton seasonal cycles that typically have

lower bloom amplitudes, are more gradual and have longer durations (Figure 2). The TS method tends to produce
earlier bloom initiations and earlier terminations in these subtropical regions (Figure A3 d-e, j-k). In these regions
the chl-a min-max range is relatively small, thus a 5% threshold may be exceeded earlier in both termination and
initiation. The RC method, based on the rate of change, is likely to produce later bloom timing dates in more
gradual blooms. There is agreement in the resultant bloom durations between the different methods, with similar
large-scale patterns being reproduced by all three methods (Figure 3c, Figure A3m-o). Unsurprisingly, in the
oligotrophic regions, differences between the methods in bloom duration do not translate to large differences in
the integrated and mean bloom chlorophyll because of the low magnitude of the chlorophyll (Figure 2a-c, Figure
3 c-e). There are however, corresponding regions with more noteworthy disagreements in both duration and mean
and integrated bloom chlorophyll, for example in the energetic regions of the Antarctic Circumpolar Current,
particularly near sub-Antarctic Islands, and localised coastal regions with significant river runoff, such as in the
Atlantic where the Amazon River discharge occurs. These areas of large STDs between the methods are driven
predominantly by the TS method (Figure A3p-r), which tends to result in shorter blooms, due to later initiations
and earlier terminations (Figure A3 d, e, j, k).

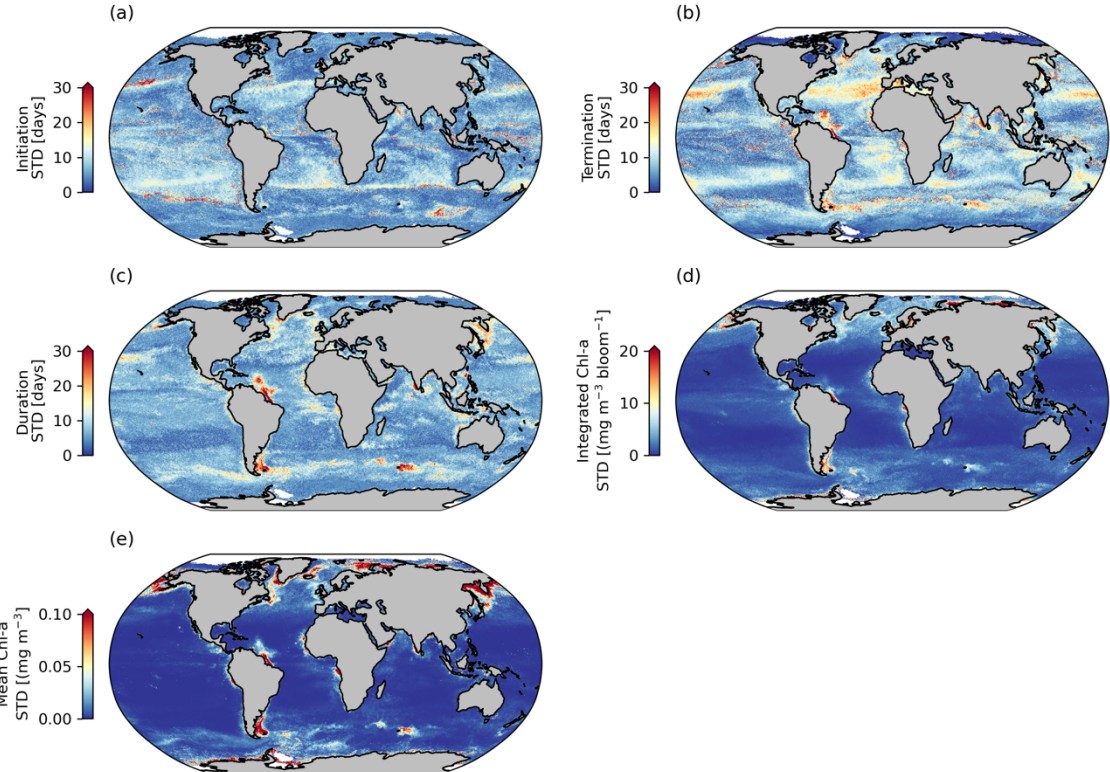

Figure 3: Comparisons between phenological detection methods. Shown are standard deviations (STD) calculated
between the biomass-based threshold method, the cumulative biomass-based threshold method and the relative of
change method , for selected seasonal phytoplankton bloom metrics, including (a) bloom initiation, (b) bloom
termination, (c) bloom duration, (d) bloom integrated chl-a and (e) bloom mean chl-a.

## 3.3. High-resolution phenology indices

The derived phenology data product presented here is offered at three different horizontal resolutions (4, 9 and 25 km), which when compared on a global scale (Figure 4) shows little to no difference in the overall mean distribution of three selected phytoplankton seasonal metrics, including bloom mean chl-a (Figure 4a), bloom duration (Figure 4b) and SCR (Figure 4c). Given that the large-scale distributions of the seasonal metrics remain virtually the same there is little benefit for the user to use the more computationally expensive 4 km product for applications across these larger scales.

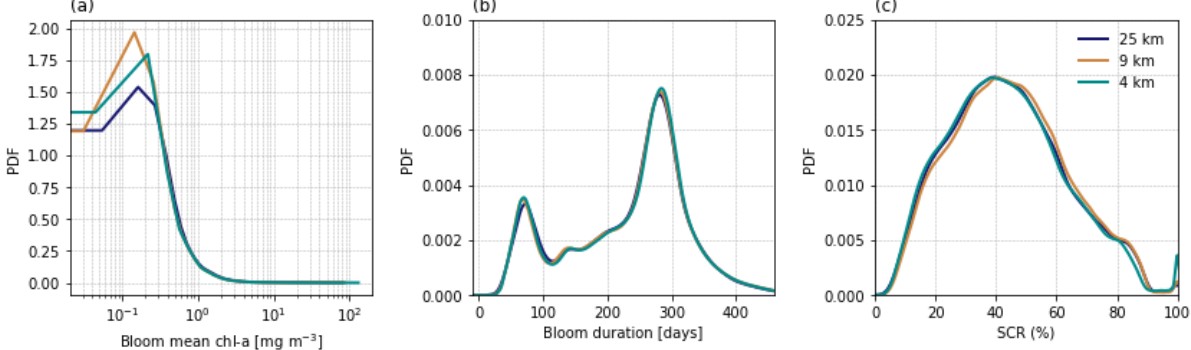

Figure 4: Probability Density Functions (PDF) of climatological mean (calculated from 1998 to 2022) phytoplankton seasonal cycle metrics, compared across three different spatial resolutions (4, 9 and 25 km) for (a) bloom mean chlorophyll-a, (b) bloom duration and (c) seasonal cycle reproducibility (SCR). The TS phenology method is used for (a) and (b).

There are, however, notable differences in the resolution of the product on smaller regional scales which appear qualitatively different when compared at two example sites (Figure 5). The sites were selected to reflect regions where a critical dependence is anticipated on the timing and magnitude of seasonal phytoplankton production. The Benguela upwelling system (Figure 5a-c), off the west coast of South Africa is an essential region for supporting key fisheries, while the subAntarctic Kerguelen Island (Figure 5d-f) is a vulnerable marine ecosystem that supports a number of key species. The coarseness of the 25 km product is clearly evident in both sites at these scales, it is considerably more pixelated and there are notable patches where there are differences in the resultant phenological metric between resolutions. For example, in the near-shore of St Helena Bay the integrated bloom chl-a climatology (2017-2022) differs between resolutions from 1654 mg m$^{-3}$ bloom$^{-1}$, 1841 mg m$^{-3}$ bloom$^{-1}$, and 1843 mg m$^{-3}$ bloom$^{-1}$, for the 25 km, 9 km and 4 km maps respectively, representing a ~10% underestimation by the 25 km product. At Kerguelen Island, the interaction of the Polar Front with shallow bathymetry generates persistent fine-scale ocean dynamics that set strong regional gradients in phytoplankton production (Park et al., 2014). These fine-scale gradients are clearly seen in the spatial variability of bloom duration captured by the higher resolution products. The 'footprint' of the island is evident in the extended bloom durations occurring over the shallow plateau associated with the island where there is considerable resuspension of dissolved iron, a key limiting nutrient (Blain et al., 2001). These examples highlight how this data product can be applied to derive valuable indicators for use in national biodiversity assessments, pelagic ecosystems mapping and marine resource

management with the added potential of monitoring change in climate sensitive regions relevant for ecosystem services. For regional studies or applications in coastal domains it is recommended that users favour the high spatial resolution product, as it could facilitate detection of finer scale delineations of phenoregions in transitional waters or detect fine scale distributions in phenology metrics that are associated with physical or oceanographic features such as eddies, bays, and upwelling cells. While some phenology indicators produced from daily data could offer additional insights into coastal regions with high temporal variability (e.g., Ferreira et al. 2021), our dataset offers a resource for areas where long gaps in the time-series could negate the use of daily data.

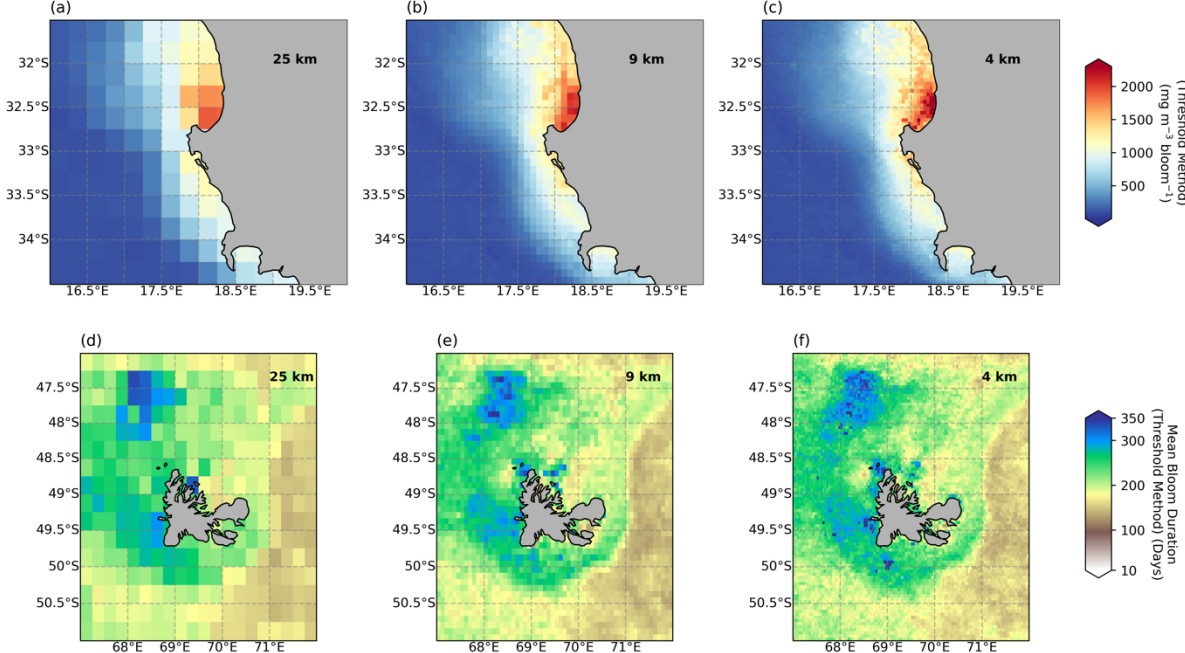

Figure 5: Regional domains comparing the impact of different resolutions (a,d) 25 km, (b,e) 9 km and (c,f) 4 km on (a-c) bloom integrated chl-a and (d-f) the bloom duration averaged from 2017-2022 for (a-c) the Benguela upwelling system off the west coast of South Africa and (d-f) Kerguelen, a subAntarctic island in the Southern Ocean.

## 4 Limitations of the phenology algorithm and future developments

The diversity of the phytoplankton seasonal cycles across the global ocean makes it challenging to generalise a single methodological approach that is capable of capturing all phenological metrics accurately. Our attempt to do so with this data product may lead to some irregularities, most notably when applied to regions with a poorly defined or unique seasonal cycle. For example, in ultra-oligotrophic regions where the bloom amplitude is particularly low and intraseasonal variability particularly high, our detection method prescribes long bloom durations that may exceed one year and can lead to overlapping bloom slices. Another example is regions with bi-modal blooms, where there is a well-defined summer and winter bloom in a given annual cycle. Although our phytoplankton phenology detection method is designed to allow for multiple peaks to occur within a bloom cycle; it has not been designed to cater for bimodal annual cycles, which would require the identification of separate summer and winter initiation and termination indices. In these instances our method may result in extended bloom durations. While these regions are relatively uncommon (e.g. Racault et al., 2017, Figure 2c), they do exist, as is

the case with the Red Sea (Racault et al. 2015). Future developments of this data product will endeavour to
incorporate updates and improvements to the detection methods to better cope with these irregularities. We
welcome users to reach out if other irregularities are identified within a specific area of interest and to work with
the authors to improve future versions of the product. All future changes to the product will be fully documented
on Zenodo as new versions are released.
**5 Data and code availability**
The data are available on the Zenodo repository under the following DOIs, 4 km: 10.5281/zenodo.8402932, 9 km:
10.5281/zenodo.8402847 and 25 km: 10.5281/zenodo.8402823 (Nicholson et al., 2023a, b, c). Chl-a data, used to
develop the phytoplankton phenology product, is available from the Ocean Colour–CCI dataset (v.6.0) at
https://esa-oceancolour-cci.org. The code used to produce the figures of this manuscript can be found
https://github.com/sarahnicholson/global_phytoplankton_phenology.
**6 Conclusions**
The satellite-derived data product presented here provides a 25-year continuous record of key phytoplankton
seasonal cycle metrics (phytoplankton bloom phenology, bloom seasonality and bloom magnitude) on a global-
scale. It includes three different phenology detection methods that are widely used by the community. We do not
advocate for a particular method over another, the strengths and weaknesses of these different approaches have
been highlighted in other studies (e.g., Brody et al., 2013), it is up to the user to choose which (if not all) is the
most appropriate for their research applications. The data product is also provided at three different horizontal
resolutions (4, 9 and 25 km) for regional versus global-scale application. This product is applicable for a broad
range of national to international research and industry applications. Its primary strength is that it can be used to
assess, monitor, and understand regional to global-scale characteristics in phytoplankton phenology and to detect
change associated with environmental drivers, which is critical for effective management of marine ecosystems
and fisheries. This data product will undergo regular updates for future applications and extended time series
analysis, which typically happens every two years. It will also be updated when data is temporally extended or
when the OC-CCI releases any version updates beyond v.6.0 that will include backwards corrections for previous
years, so the entire dataset aligns with the latest version of OC-CCI. This helps to prevent the retention of
erroneous values within the data set.
**Appendix A**

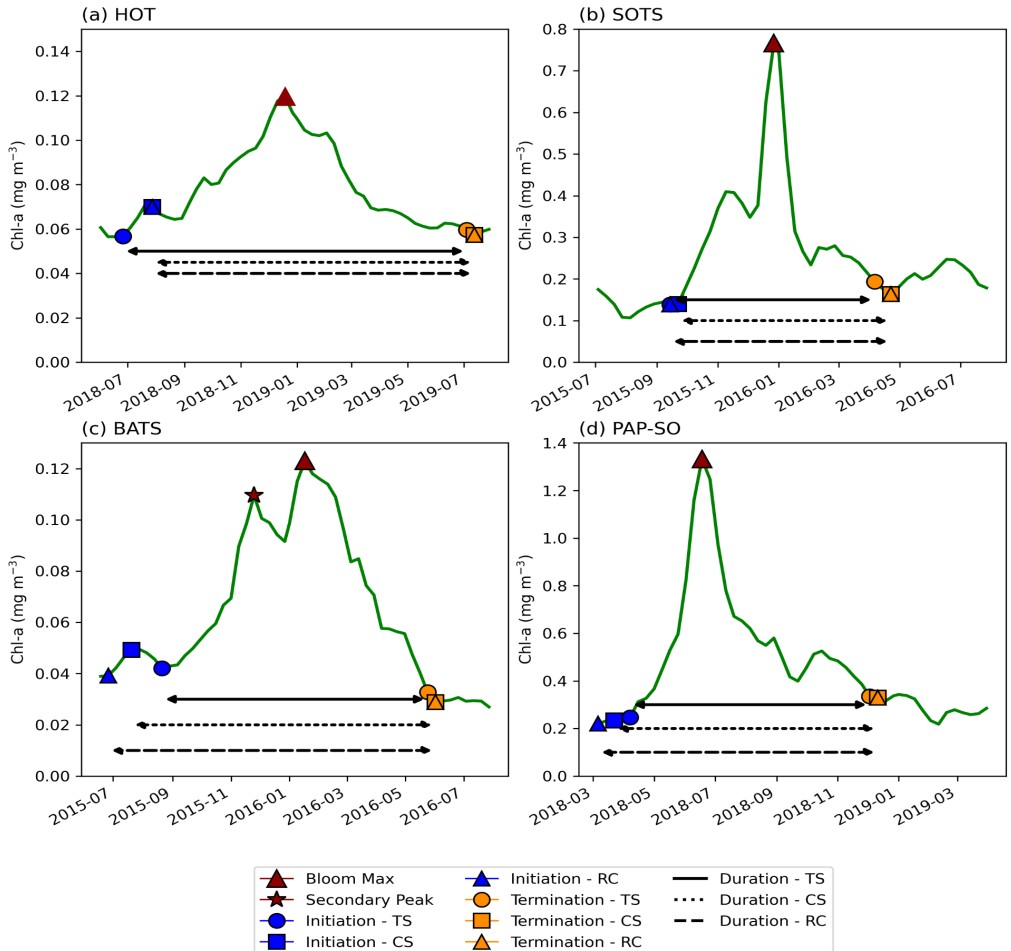

Figure A1: Examples of phytoplankton bloom seasonal cycles of satellite-derived chlorophyll-a from OC-CCI and comparisons in phenological detection methods at key sustained observing stations across the global ocean. For (a) Hawaii Ocean Time-series (HOT, 21° 20.6'N, 158° 16.4'W), (b) Southern Ocean Time Series Observatory (SOTS, 140°E, 47°S), (c) Bermuda Atlantic Time-series Study (BATS, 31° 50' N, 64° 10'W) and (d) Porcupine Abyssal Plain (PAP-SO, 49°N, 16.5°W) sustained observatory time-series.

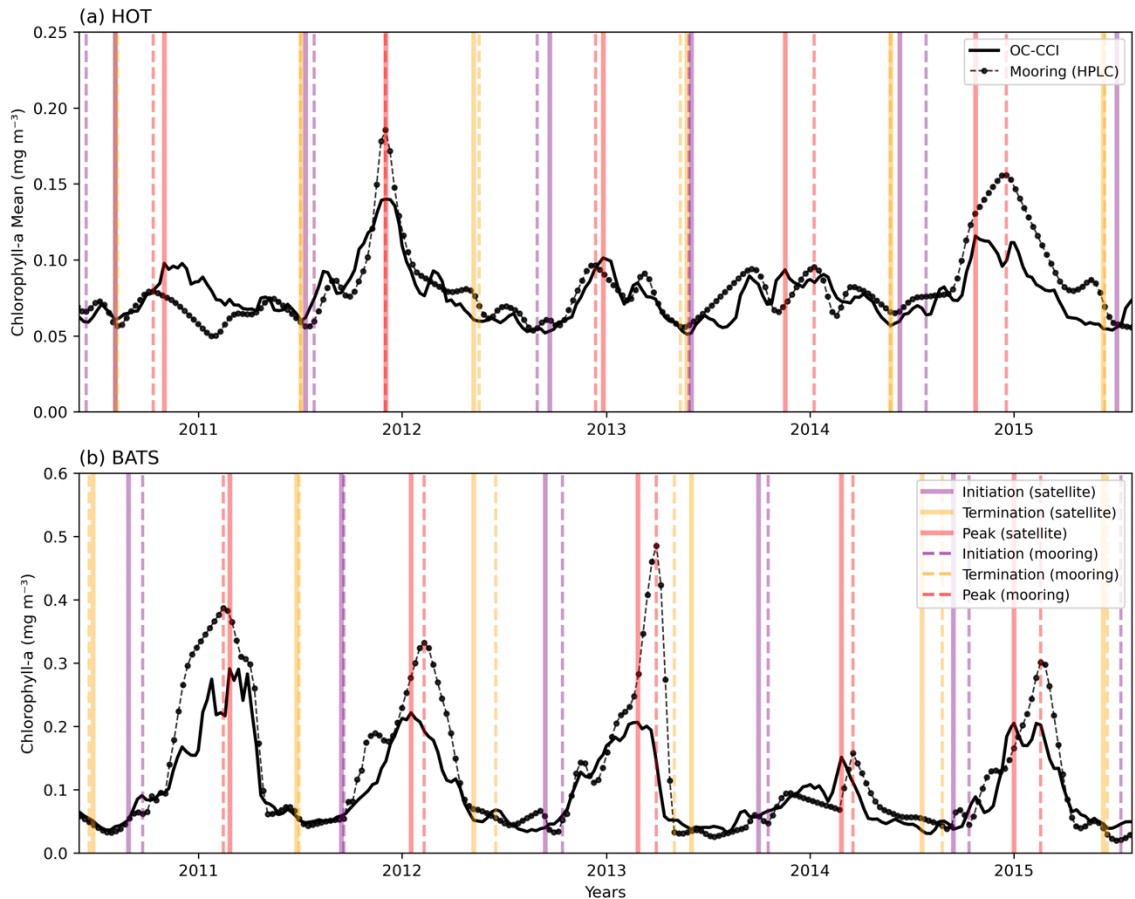


Figure A2: Comparison of five years of in situ chlorophyll-a measurements (Valente et al. 2022) with satellite-
derived chlorophyll-a (OC-CCI), along with key phenological indices (solid and dashed vertical lines for satellite
and mooring, respectively) at two sustained observing stations: (a) Hawaii Ocean Time-Series (HOT, 21° 20.6'N,
158° 16.4'W) and (b) Bermuda Atlantic Time-Series Study (BATS, 31° 50'N, 64° 10'W).

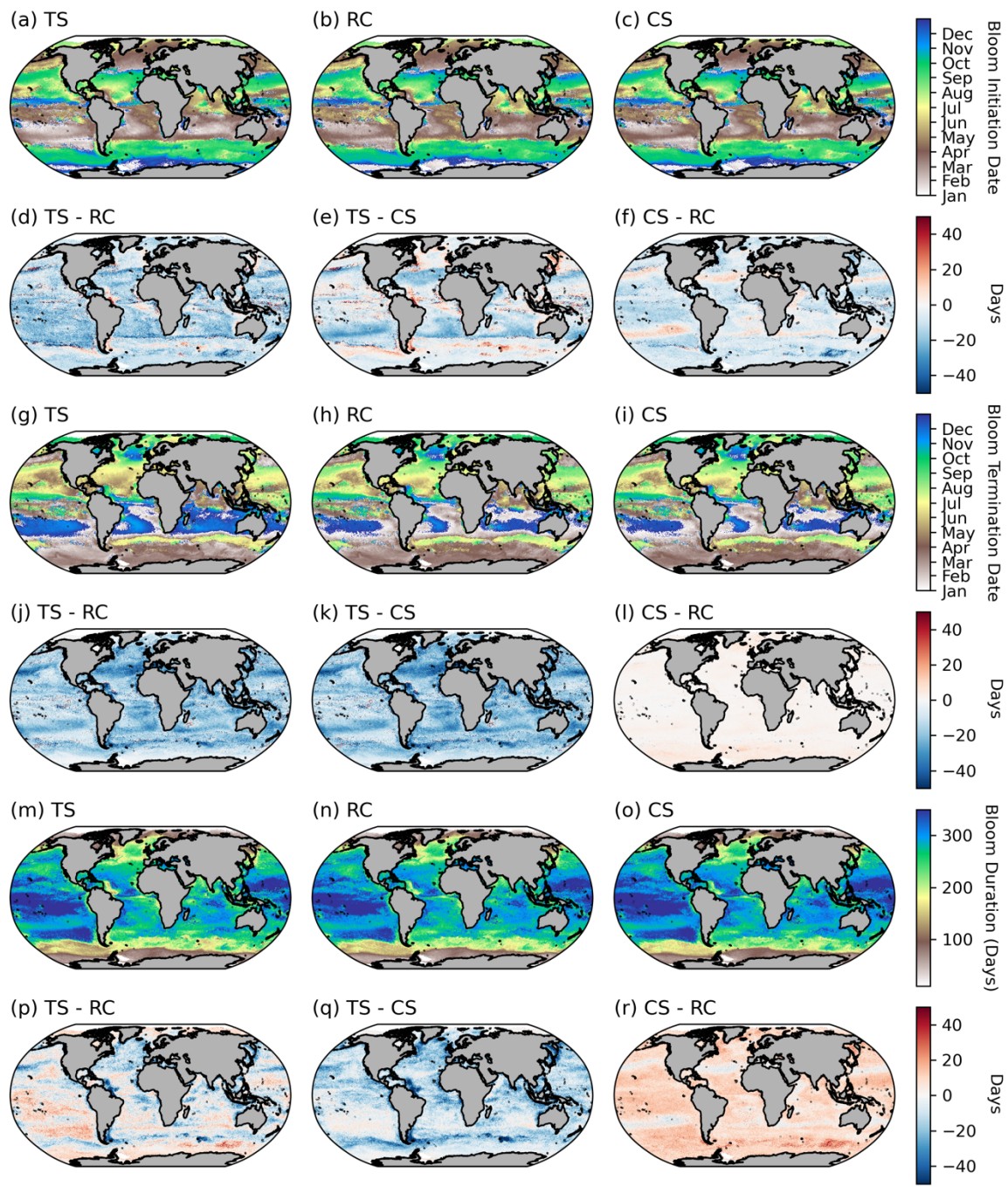


Figure A3. Comparisons between phenological detection methods. The climatological means [1998 - 2022] for (a-c) bloom initiation, (g-i) bloom termination, and (m-o) bloom duration. The differences between the climatological means for the biomass-based threshold method (TS), the cumulative biomass-based threshold method (CS) and the rate of change method (RC) are provided for bloom initiation (d-f), bloom termination (j-l) and bloom duration (p-r).


**Author contributions. Conceptualization: SN, TJRK, SJT. Formal analysis: SN, TJRK, MES, Software:**
**TRJK, SN, NC. Visualisations: SN, TJRK. Writing – original draft: SN. Writing, reviewing, and editing:**
**SN, TJRK, SJT, MES, NC.**
**Competing interests. The contact author has declared that none of the authors has any competing interests.**
**Acknowledgements**
We would like to acknowledge the OC-CCI group for providing the satellite data used in this manuscript. The
authors acknowledge their institutional support from the CSIR Parliamentary Grant (0000005278) and the
Department of Science and Innovation. We similarly acknowledge the Centre for High-Performance Computing
(NICIS-CHPC) for the support and computational hours required for the analysis of this work. SN, TRK, ST and
NC acknowledge the National Research Foundation (SANAP200324510487; SANAP200511521175;
MCR210429598142).

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
