# Peer review of "Satellite-derived global ocean phytoplankton phenology"

_Earth System Science Data, 2024_

## Referee Comment (RC1)

**Review of manuscript essd-2024-21**

**"Observed global ocean phytoplankton 1 phenology indices."**

This manuscript provides satellite-derived chlorophyll-a data from the Ocean Colour Climate Change Initiative at 4, 9, and 25 km spatial resolutions. The dataset is valuable and can be used in a wide range of research topics and real applications. A few concerns are listed below and the 1 one is the major one.

**1. The accuracy of the data in reflecting the phytoplankton phenology has not been stated with field observation. This is needed to increase the confidence of the satellite-based data. I suggest the author choose a few typic locations with some field survey data and compare the direct observation with your data.**

2. In the method, a few critical percentage values were used to determine the phytoplankton phenology parameters, e.g., 75% of the amplitude of the bloom maximum peak magnitude, 5% of the chl-a range, 15% of the total cumulative chi-a concentration and of the median rate of change in chl-a concentration. Why do you choose different values as critical points and are there any standards or citations that suggest the use of these values? The reason should be pointed out in the manuscript. For example, three peaks would be detected in Figure 1 if a lower critical value is used.

3. Lines 210 to 214, it is not very clear to me, please re-edit to make your idea more clear.

4. Line 260, please describe the explanation provided in Brody et al., (2013) shortly here, so that the readers could understand the reason easier and more directly.

5. Figure 3 and the relevant text compare the three detection methods using CoV values, it is not clear if the three methods all differ from each other or only one of them resulted in descripency when CoV is large. A conclusive sentence is needed in the text. For example, the comparison between three values in line 312 is very clear that the integrated bloom chl-a climatology (2017-2022) is similar using 9 and 4 km maps, but is different from that determined using a 25 km map.

---

## Author Comment (AC1)

We thank the reviewers for their insightful comments and time taken to review our manuscript. I apologise for the considerable time that has passed since this first round of revisions, I have recently returned back to work from maternity leave. Please see the responses below in blue.

"Observed global ocean phytoplankton phenology indices."

**Reviewer 1**

This manuscript provides satellite-derived chlorophyll-a data from the Ocean Colour Climate Change Initiative at 4, 9, and 25 km spatial resolutions. The dataset is valuable and can be used in a wide range of research topics and real applications. A few concerns are listed below and the 1 one is the major one.

**1. The accuracy of the data in reflecting the phytoplankton phenology has not been stated with field observation. This is needed to increase the confidence of the satellite-based data. I suggest the author choose a few typic locations with some field survey data and compare the direct observation with your data.**

We thank the reviewer for the comment regarding the accuracy and validation of the underlying data used to generate the phenological indices presented here. This is indeed crucial for ensuring the reliability of the phenological indices. However, we feel it is out of the scope of this study to conduct an independent validation of the OC-CCI data product. This has been done extensively by the European Space Agency (ESA), Ocean Color Climate Change Initiative Team and provided by version 6 documentation (https://docs.pml.space/share/s/fzNSPb4aQaSDvO7xBNOClw) and also refer to details in publication: https://www.mdpi.com/1424-8220/19/19/4285
For example, see below figure comparing OC-CCI data when matched against the CCI database of in situ data, extracted from OC-CCI v6 documentation (see section 5 from https://docs.pml.space/share/s/fzNSPb4aQaSDvO7xBNOClw.)

[Figure]

The OC-CCI data are validated, error-characterised Essential Climate Variable products that are widely used and accepted within the scientific community for a range of applications

(refer to list of published papers: https://climate.esa.int/en/projects/ocean-colour/) that include studies on phytoplankton phenology, primary productivity, and biogeochemical cycles. The dataset's reliability is thus already well-recognized, and numerous studies have utilised OC-CCI data without conducting independent validation, relying instead on the established credibility of the ESA's validation efforts. We note that there are already several independent regional validation efforts of this product for phytoplankton phenology, we select two examples for the tropical ocean https://www.nature.com/articles/s41598-018-37370-4 and from the subantarctic Southern Ocean: https://doi.org/10.1093/icesjms/fsv105. Nevertheless, we do also recognise that these remote sensing satellite products are not perfect and while regionally variable amplitude biases exist, the OC-CCI product was generated with the specific intent to be used for global application to generate long-term system change studies, such as phytoplankton dynamics under climate variability and change. In many such instances, even if there is a regional bias in satellite estimates relative to in situ chlorophyll concentrations, the value of satellite application remains critical for evaluating multi-decadal changes in the characteristics of the seasonal cycle (e.g. in phenology, variability and magnitude) that reflect a biological response to changes in physico chemical conditions. (refer to https://www.mdpi.com/1424-8220/19/19/4285).

That said, we have modified the text of the manuscript to recognise the existence of well documented limitations of satellite remote sensing data products (e.g. regional biases) but also to reflect on efforts made by OC-CCI to generate a data product of value for generating climatologies and assessing variability and change. Refer to additional text for example in the introduction, see italics below, from line 112-127:

> *"We note however that despite their obvious spatial and temporal advantages, remotely detected water-leaving radiances emanate from only the first optical depth, and give little quantitative information about the vertical structure of the water column, which can be particularly important in low nutrient regions where a subsurface chl-a maxima is prevalent. In addition, we recognise that the OC-CCI chl-a data product may exhibit regional biases (that can vary in both magnitude and direction) and arise from several factors inherent to both satellite remote sensing technology and the complexities of ocean ecosystems. One example is that algorithms are often regionally trained on datasets from specific parts of the world, which can result in discrepancies when applied globally. Despite these regional biases, satellite ocean colour chl-a data products remain highly valuable, especially when the goal is to identify patterns in the seasonal cycle of phytoplankton and how these patterns evolve over time. While local accuracy may be impacted by biases, the broader trends—such as the timing of spring blooms, the intensity of summer productivity, or the length of growing season—are still well captured. This is because biases tend to be relatively consistent over time in any given region, allowing researchers to focus on changes in these patterns rather than on the absolute values. These long-term changes in the seasonal cycle are crucial for understanding how marine ecosystems respond to environmental stressors like warming temperatures, ocean acidification, and changes in nutrient availability."*

And to the methodology, refer to additional text in italics, on lines 170-176:

> "This version marks a substantial change to previous versions (e.g., v5.0, see Sathyendranath et al., 2021) in that it incorporates Sentinel 3B OLCI data, the MERIS-4[th] reprocessing dataset, upgraded Quasi-Analytical algorithm (QAAv6) and

the exclusion of MODIS and VIIRS data after 2019 (refer to D4.2 - Product User Guide for v6.0 Dataset from https://climate.esa.int/en/projects/ocean-colour/key-documents/ for further details on processing and validation). The OC-CCI observational product was generated with the specific aim of studying phytoplankton dynamics at seasonal to interannual scales. Indeed, it has been used widely by the scientific community for studying phytoplankton phenology (e.g. Ferreira et al., 2021; Gittings et al., 2019, 2021; Racault et al., 2017; Thomalla et al., 2015, 2023). "

2. In the method, a few critical percentage values were used to determine the phytoplankton phenology parameters, e.g., 75% of the amplitude of the bloom maximum peak magnitude, 5% of the chl-a range, 15% of the total cumulative chi-a concentration and of the median rate of change in chl-a concentration. Why do you choose different values as critical points and are there any standards or citations that suggest the use of these values? The reason should be pointed out in the manuscript. For example, three peaks would be detected in Figure 1 if a lower critical value is used

The 5% of the chl-a range, 15% of the total cumulative chi-a concentration and of the median rate of change in chl-a concentration are in accordance with previous literature. The sensitivities of the different percentages chosen have been explored in these other studies already. E.g. see excerpt from Brody et al. 2013: "*The 15% of the cumulative biomass threshold we used produced BSDs more closely aligned with the first increases in chlorophyll biomass, while a 30% threshold would produce BSDs associated with the largest increases in biomass*" and "*We determined that a threshold of 15% of the total biomass (cross) best predicted the bloom initiation date for our study area using two techniques. First, we visually inspected multiple chlorophyll time series over the entire study area with BSDs determined using six thresholds (5–30%). From these time series, we found that thresholds of 10–15% best predicted the bloom initiation date in subtropical regions, while threshold of 15–20% best predicted the bloom initiation date in subpolar regions. We then plotted, for the six thresholds, the number of occurrences in which chlorophyll levels at each threshold's BSD exceeded chlorophyll levels prior to the BSD but were not larger than the yearly chlorophyll median plus one standard deviation. We found that the 15% threshold had the largest number of points with increasing chlorophyll levels at the BSD, which confirmed the results of the visual inspection and led to the choice of the 15% threshold.*"

We have cited the relevant publications where these thresholds are first mentioned in our study:

*Lines 277-280:*
> "*To note, the above percentage thresholds are in accordance with those used by previous phenological detection studies (e.g. Ji et al. 2010, Brody et al., (2013); Hopkins et al., (2015); Thomalla et al., (2011, 2015) and Henson et al., (2018)).*"

The 75% of the amplitude of the bloom maximum peak magnitude, which was used to identify the presence of multiple bloom peaks, was chosen with the objective of identifying well defined peaks that were similar in magnitude (and within a given range of time) from the main peak. A threshold higher than that ends up not meeting the purpose for counting multiple peaks, whilst a threshold lower than this is catching subseasonal variability. The

percentage threshold was chosen over a magnitude based threshold as it remains robust in regions with higher or lower chlorophyll values.

We have added the above explanation into the manuscript, lines 250-255:
*"The 75% threshold was chosen to identify peaks with similar magnitude to the bloom maximum peak so as to allow for the occurrence of a multiple peak growing season. Choosing a threshold higher than this would likely exclude recognisable bloom peaks (which could lead to an underestimate of the bloom duration), while choosing a lower threshold may include sub-seasonal variability and lead to an overestimation of the bloom duration"*

3. Lines 210 to 214, it is not very clear to me, please re-edit to make your idea more clear.

We have edited the relevant lines to be clearer as follows:
*"The cyclical nature of the year day calendar presents a significant challenge when calculating means of phenological indices. For example, we need to avoid a situation where the mean bloom initiation between a year with a bloom in December (day of year = 340) and a year with a bloom in January (day of year = 10) is incorrectly calculated as an average bloom initiation date in July (day of year = 175). To address this, as similarly applied in Thomalla et al. 2023, we used the Python SciPy function circmean, which calculates circular means for samples within a specified range, correctly identifying the mean as day of year 357."*

4. Line 260, please describe the explanation provided in Brody et al., (2013) shortly here, so that the readers could understand the reason easier and more directly.

We have edited the text from line 260 to include the explanation from Brody et al., 2013:
*"For example, as explained in Brody et al, (2013), the TS method which is based on the range of bloom amplitude (refer to methods) will capture the bloom start dates at the largest increase in chlorophyll concentrations. It is thus more suitable for studies wanting to investigate the match or mismatch between phytoplankton and upper trophic levels as the match-mismatch hypothesis is based on the timing of the high phytoplankton biomass period [Cushing, 1959]."*

5. Figure 3 and the relevant text compare the three detection methods using CoV values, it is not clear if the three methods all differ from each other or only one of them resulted in descripency when CoV is large. A conclusive sentence is needed in the text. For example, the comparison between three values in line 312 is very clear that the integrated bloom chl-a climatology (2017-2022) is similar using 9 and 4 km maps, but is different from that determined using a 25 km map.

We thank the reviewer for raising this important point. We have decided to change Figure 3 to the standard deviation of the climatological means of the three methods, which allows an easier interpretation of the magnitude of the differences between the methods.

[Figure]

*Figure 3: Comparisons between phenological detection methods. Shown are standard deviations (STD) calculated between the biomass-based threshold method, the cumulative biomass-based threshold method, and the relative of change method, for selected seasonal phytoplankton bloom metrics, including (a) bloom initiation, (b) bloom termination, (c) bloom duration, (d) bloom integrated chl-a and (e) bloom mean chl-a. .*

We have also included an additional figure in the Appendices which provides the climatological means of each method and the differences between them (e.g. TS - RC, TS-CS and CS-RC). This allows the user to have a better understanding of the magnitude of the differences between the methods globally.

[Figure]

*Figure A2. Comparisons between phenological detection methods. The climatological means [1998 - 2022] for (a-c) bloom initiation, (g-i) bloom termination, and (m-o) bloom duration. The differences between the climatological means for the biomass-based threshold method (TS), the cumulative biomass-based threshold method (CS) and the rate of change method (RC) are provided for bloom initiation (d-f), bloom termination (j-l) and bloom duration (p-r).*

The text of section "3.2 Comparisons of phenology detection methods" has thus been substantially revised to reflect the changes to the figures. We have also included a bit more discussion around the disagreements of the methods. For example, refer to added text in section 3.2

*"In general, there is stronger agreement between methods in the higher subpolar latitudes compared to subtropical latitudes, as evidenced by slightly elevated STDs in the subtropical gyres (Figure 3a,b). The subtropical oligotrophic regions are characterised by phytoplankton seasonal cycles that typically have lower bloom amplitudes, are more gradual and have longer duration (Figure 2). The TS method tends to produce earlier bloom initiations and earlier terminations in these subtropical regions (Figure A2 d-e, j-k). In these regions the chl-a min-max range is relatively small, thus a 5% threshold may be exceeded earlier in both termination and initiation. While, on the other hand the RC method, based on the rate of change, is likely to produce later bloom timings dates in more gradual blooms"*

*"Unsurprisingly, in the oligotrophic regions, differences between the methods in bloom duration do not translate to large differences in the integrated and mean bloom chlorophyll because of the low magnitude of the chlorophyll (Figure 1a-c, Figure 3 c-e). There are, however, corresponding regions of larger disagreements in duration and mean and integrated bloom chlorophyll, for example in energetic regions of the Antarctic Circumpolar Current, particularly near sub-Antarctic Islands, and localised coastal regions with significant river runoff, such as in the Atlantic where the Amazon River discharge occurs. These areas of large STDs between the methods are driven predominantly by the TS method (Figure A2p-r), which tends to result in shorter blooms, due to later initiations and earlier terminations (Figure A2 d, e, in these above-mentioned locations.
"*

**Reviewer 2**

This work documents the process of creation of a dataset of phenological indexes of phytoplankton blooms on the global ocean, using the 26-years timeseries of the satellite-derived chlorophyll gathered by the OCCCI. The dataset could be potentially useful to feed other analysis. However, in its current state I see two main weaknesses, it is not validated (1), and it does not provide error estimates (2).

General comments

1. The manuscript shows, technically speaking, a great data analysis. The authors have done a rigorous job collecting data, filling gaps and applying globally-appropriate bloom detection methods. Their results look very neat, and it would be very interesting to see a deeper analysis. However, I do not see so clear the potential of these data being useful in the future to other scientists and therefore being published as an ESSD dataset.

We thank the reviewer for this critical feedback with regards to the usefulness of the global multi-year product of phytoplankton phenology presented here. There are a number of reasons why these data are useful for scientists and other practitioners, which we reiterate here. Phytoplankton form the base of the marine food web, they are highly sensitive to changes in physico-chemical conditions, such that long-term phenology datasets can help understand how phytoplankton are responding to environmental changes (sea temperature, ice cover, light availability, nutrients etc.). These changes in phenology may have cascading impacts on the entire ecosystems (fish populations, mammals and birds), thus monitoring

phenological changes can provide early warnings of ecosystem shifts or decline in ecosystem health. Phenology data can be used to inform fisheries scientists and management by predicting the timing and extent of primary production, which supports fish stocks. Understanding phytoplankton phenology can help improve earth system models with respect to carbon sequestration and nutrient cycling, this is crucial for predicting future changes in marine environments and for making informed policy decisions. It is also important to recognise that the generation and continuation of this product is extremely computationally expensive. Not everyone (or country) has the access to compute resources required to generate this product to support their studies and/or to use in ecosystem management, particularly those from the global south where access to HPC resources can be limited.

We hope that the modifications and below additional text incorporated into the introduction of the manuscript will help to clarify the value of such a data set to multiple users. Refer to italics text for additions:

Lines 93-96:

*"Having access to a global data product that characterises the seasonal cycle of phytoplankton over the last 25 years and into the future can thus provide a valuable tool to users that require an understanding of key aspects of the growing season and how these may be changing over time."*

*Lines 125-127:*

*"These long-term changes in the seasonal cycle are crucial for understanding how marine ecosystems respond to environmental stressors like warming temperatures, ocean acidification, and changes in nutrient availability."*

*Lines 150-154 , refer to italic text:*

In addition, a phenology data product such as this can provide a useful aid for the planning of oceanographic research campaigns that *wish to align with or determine their occupation relative to key aspects of the growing season. Finally, this derived observational data product could also be valuable to support those users without the programming know-how or access to computationally expensive resources that are required to generate it*

*Lines 158-159:*

*The data product facilitates the global characterisation of the climatological seasonal cycle and can be used to identify the sensitivity of the seasonal cycle to change (through the analysis of trends and anomalies).*

The OCCCI Chl-a is itself a satellite-derived product, based on the disaggregation of the world ocean in a certain number of optically-homogeneous water classes. However, global algorithms, even blending different waterclasses, do not compare necessarily well to observations in certain regions, where regional algorithms are proposed (e.g. Johnson *et al*. 2013 in the Southern Ocean [https://doi.org/10.1002/jgrc.20270]; Volpe *et al*. 2019 in the

Mediterranean Sea [https://doi.org/10.5194/os-15-127-2019]). To the best of my knowledge OCCCI do not consider regional-specific algorithms.

And on top of that it is the uncertainty of the phenological analysis performed (which is also not very well documented, see my next comment). With such level of derivation I do not see how these metrics provided could be considered observed data. This issue could be overcome if the authors present some comparison to in situ observations of Chl-a timeseries, observed phenology or other common standards, but that is not done in the current version. Have the different bloom detection methods been validated with observational data on their own? Since obtaining global-scale validation data could be challenging, maybe one option is to perform a more formal analysis on the agreement/disagreement among methods considering in which temporal/spatial domain they have been validated independently.

We appreciate the reviewer's comments regarding the use of the OC-CCI data in our phenological analysis. We fully acknowledge that the OC-CCI Chl-a is a satellite-derived product and that its derivation is calculated by blending algorithms based on optical water type and that regional biases may still occur. However, in the absence of a single unifying algorithm applicable to all optical satellite sensors and across all marine environments, the optical water type methodological framework used by the OC-CCI currently provides the only viable solution for dealing with long-term, multi-sensor datasets at a global scale. While we understand that global algorithms may not always align perfectly with regional observations, the OC-CCI have put considerable effort into defining the per-pixel and regional uncertainties associated with each of their products, as well as highlighting currently under-represented regions (e.g. regions with lower cumulative water class membership in https://www.sciencedirect.com/science/article/pii/S0034425717301396#f0025 ). We would also note that regional validation assessments do not always agree, with outcomes that are strongly dependant on the strictness of the match up criteria being implemented (in both time and space) and the methodology being used to determine the in situ concentrations used in the comparison (e.g. HPLC versus acetone extracted fluorometric chlorophyll). As an example, we refer the reviewer to https://www.mdpi.com/2072-4292/11/15/1793. Nevertheless, even if susceptible to regional bias, this does not make the application of a satellite remote sensing data product inappropriate for characterising the phenology of the climatological seasonal cycle or the characteristics of variability (e.g. Thomalla et al., 2011, Tang et al,. 2021, Hauko et al. 2021) or to detect trends in any of these seasonal metrics (e.g. Silva et al. 2021, Anjaneyan et al. 2023, Delgado et al. 2023, Thomalla et al., 2023).

We have added the following text into the introduction which highlights some of these limitations, see italics below, from line 112-127:

> *"We note however that despite their obvious spatial and temporal advantages, remotely detected water-leaving radiances emanate from only the first optical depth, and give little quantitative information about the vertical structure of the water column, which can be particularly important in low nutrient regions where a subsurface chl-a maxima is prevalent. In addition, we recognise that the OC-CCI chl-a data product may exhibit regional biases (that can vary in both magnitude and direction) and arise from several factors inherent to both satellite remote sensing technology and the complexities of ocean ecosystems. One example is that algorithms are often regionally trained on datasets from specific parts of the world, which can result in discrepancies*

*when applied globally. Despite these regional biases, satellite ocean colour chl-a data products remain highly valuable, especially when the goal is to identify patterns in the seasonal cycle of phytoplankton and how these patterns evolve over time. While local accuracy may be impacted by biases, the broader trends—such as the timing of spring blooms, the intensity of summer productivity, or the length of growing season—are still well captured. This is because biases tend to be relatively consistent over time in any given region, allowing researchers to focus on changes in these patterns rather than on the absolute values. These long-term changes in the seasonal cycle are crucial for understanding how marine ecosystems respond to environmental stressors like warming temperatures, ocean acidification, and changes in nutrient availability."*

With regards to concerns around the validation of OC-CCI satellite observations with independant in situ measurements, we refer the reviewer to the first response to Reviewer 1 above, which highlights the credibility of ESA's validation efforts and the numerous studies that have utilised OC-CCI data without conducting independent validation, including for applications that determine bloom phenology. It is similarly beyond the scope of this data product submission to globally validate the different bloom detection methods. Suffice to say that these different methods are well documented and have been applied over many years to different data streams (e.g. chlorophyll from satellites, gliders, floats, moorings) and in many different ocean regions (Ji et al. 2010, Racault et al. 2012, 2014, Brody et al. 2013, Thomalla et al. 2015, 2023, Gittings et al. 2019, 2021, Ferreira et al. 2021, Silva et al. 2021). We do however recognise that different methods are more or less sensitive to different characteristics of the time series (e.g. to rapid adjustments in rates of change or inflection points etc.) such that they have a tendency to select different dates for initiation/ termination etc. It is for this reason that we apply three different (well documented) approaches for detecting bloom phenology. This allows the user the ability to assess the similarities or differences between different detection methods for their region of interest. Applying all three methods allows the user to discern which method is most appropriate to their region OR to use all three methods to  provide a range of variability for each bloom metric. We have provided more discussion on where and when the different methods disagree on a global scale and by how much. We have revised Figure 3 and included an additional Figure A2 added to Appendices, and also request that you please refer to our response to comment 5 from Reviewer 1 above.

Regarding the comment on whether these phenological metrics should even be considered observed data. It is our understanding that satellite data are generally considered a remote sensing observation. However, we recognise the reviewers' concerns that having undergone so much processing these measurements could easily be considered a derived product. In an attempt to address this concern the first time we use OC-CCI satellite remote sensing we now refer to it as an "observational data product", while for the phenology metrics that we derive from this observational product we now refer to them as "derived observational data products".

2. The quality of the presentation is high. The dataset is accessible and straightforward to interpret. However, another big concern is that the dataset does not include any estimate of error associated to the metrics given. It is of utmost importance to provide such an error, considering that the trends on such metrics seem to be on the range of 5-10 days per

decade. Dispersion metrics around the mean for each pixel (in the 9km and 25km versions) are also missing. I think these can be provided since phenology indexes are computed in the 4km version and later regridded (L115). There is no discussion about the potential sources of errors and limitations of the bloom detection methods, only references to other works (L135). Maybe the authors could mention the potential caveats of the methods when they elaborate on the agreement/disagreement between methods (L267).

We thank the reviewer for raising this important concern regarding the absence of error estimates in the phenological metrics derived from the OC-CCI chl-a dataset. While it is challenging to provide meaningful error estimates for the phenological timing metrics generated from the OC-CCI observational product, we do recognise the importance of including the uncertainty, which will support more reliable results and ultimately strengthens the data product towards more informed decision making regarding the management and preservation of ocean ecosystems (see Werther et al. 2023).

There are multiple approaches available to address the uncertainty in a phenological data product. In our case, we purposefully chose to provide users with three different detection methods, each applied to three resolutions: the native 4 km and regridded 9 km and 25 km. This results in nine unique realisations for each phenological metric for any given region globally. This was done in response to recognising that the detection methods each have their strengths and weaknesses and that the regridding of chlorophyll-a data introduces additional variability. These factors cause some regional differences in the phenological outcomes, giving the users insight into variability stemming from methodological and resolution choices.

Another approach could be to explore the uncertainties associated with the underlying chlorophyll data product. The OC-CCI observational product provides per pixel "uncertainty" metrics for chl-a (refer to product manual https://docs.pml.space/share/s/fzNSPb4aQaSDvO7xBNOClw and Jackson et al,. 2017). The uncertainty metrics, bias and RMSD, are calculated for each optical water type derived from match ups between in situ and satellite derived chl-a. These class statistics are then used to compute daily per-pixel uncertainty. However, it is important to note, that these uncertainty metrics are only as good as the in situ data available. For example, we note that there is not an even spread across all waterclass types, meaning that the weighted errors themselves likely have implicit biases (Sathyendranath et al. 2019). Nevertheless, to incorporate the uncertainty metrics from OC-CCI into the derived phenological observational product, we propose to calculate the standard deviation for the per pixel chlorophyll-a time-series that is used to calculate the phenology, following:

STD = $\sqrt{RMSD^2 - bias^2}$

This gives an uncertainty band around the chlorophyll values provided by OC-CCI. For example, if at a specific time (e.g. the seasonal bloom peak) the chlorophyll concentration is 1.5 mg/m³ and the STD is 0.1 mg/m³, the concentration might range from 1.4 to 1.6 mg/m³ . Adding and subtracting the STD to the chlorophyll time-series is done as follows (taking into account that the uncertainty metrics are log transformed):

Chl+STD = 10**(log10(chlorophyll-a)  + STD_chlorophyll-a)

Chl-STD = 10**(log10(chlorophyll-a) - STD_chlorophyll-a)

An example of the resultant lower and upper chlorophyll-a ranges (dashed-grey) are compared against the original chlorophyll-a time-series (black) for two selected pixels from the Southern Ocean Time Series Observatory (SOTS, 140°E, 47°S) and Porcupine Abyssal Plain (PAP-SO, 49°N, 16.5°W) in the below figure.

[Figure]

The phenology can then be computed on each of these time-series. For example, applying the three different phenology detection methods (TS= biomass threshold, CS = cumulative sum and RC = rate of change method) for initiation on the SOTS time-series on the 4 km dataset above, results in:

- **Chlorophyll**:  TS = 2019-09-14, CS = 2019-10-16, RC = 2019-10-16
- **Chlorophyll -  STD**:  TS, 2019-09-14, CS=2019-10-16, RC = 2019-10-16
- **Chlorophyll +  STD:**  TS, 2019-09-14, CS=2019-10-24, RC = 2019-10-16

While most of the methods produce the exact same initiation for each time-series (chl and chl±STD), the cumulative sum (CS) method, which aggregates chlorophyll values, is more sensitive to the upper-bound (+STD) chlorophyll-a values, resulting in a different initiation date for the chlorophyll+STD. As such, it is anticipated that the phenological timing uncertainties will be small. We note however that the reported uncertainties for bulk metrics (i.e. bloom amplitude, bloom mean and bloom integrated chl-a) could be large.

The reprocessing of the 4 km, 9 km and 25 km phenology datasets for the three different methods, for three chlorophyll-a time-series (as in the figure above: chlorophyll-a and chlorophyll-a ± STD) is not a trivial task. It will require an enormous amount of compute resources, effort and time. As such, although we are unable to have this complete in time for this review we will endeavour, as feasibly possible (e.g. dependant on access to the required

compute resources), to release an updated version of the dataset that includes these additional uncertainties on acceptance of the manuscript. Therefore, it is anticipated that for every region globally there will be 27 realisations of each phenology indices based on i) three methods of detection, ii) three grid resolutions and iii) uncertainties in the underlying chlorophyll-a product (chlorophyll-a, chlorophyll-a+STD, chlorophyll-a - STD). We hope that the commitment to propagate these uncertainties to complement the derived data product will satisfy the reviewer and if so, we kindly request the additional time needed to implement this request in an updated version of the dataset, which we anticipate will be achievable before the end of the calendar year. We are also open to alternate suggestions of how best to propagate error uncertainties for this particular data product.

Regarding the requested dispersion metrics for bloom phenology indices, it is our understanding that this is not possible in this instance because the remotely sensed chlorophyll data is first regridded from the 4 km to the 9 km and 25 km resolution and then the phenological metrics are calculated from the regridded chlorophyll data. See excerpt: "*The phenological indices described below are calculated using three horizontal resolutions in surface chl-a, the native 4 km resolution as provided by OC-CCI and a regridded 9 km and 25 km horizontal resolution.* " Thus, we cannot do dispersion metrics as the phenology data is calculated from the gridded chlorophyll, and we end up with one value (e.g. the date of bloom initiation) per pixel (4 km, 9 km or 25 km) per year. However, the dispersion metrics for the different methods for a given resolution, as shown by standard deviation of methods for the 25 km resolution as in updated Figure 3, can easily be produced by the user.

**References noted above:**

Anjaneyan  et al. (2023) Spatio-temporal changes of winter and spring phytoplankton blooms in Arabian sea during the period 1997–2020 https://doi.org/10.1016/j.jenvman.2023.117435

Delgado, A. L., Hernández-Carrasco, I., Combes, V., Font-Muñoz, J., Pratolongo, P. D., & Basterretxea, G. (2023). Patterns and trends in Chlorophyll-a concentration and phytoplankton phenology in the biogeographical regions of Southwestern Atlantic. *Journal of Geophysical Research: Oceans*, 128, e2023JC019865. https://doi.org/10.1029/2023JC019865

Ferreira, A., Brotas, V., Palma, C., Borges, C., and Brito, A. C.: Assessing Phytoplankton Bloom Phenology in Upwelling-Influenced Regions Using Ocean Color Remote Sensing, Remote Sensing 2021, Vol. 13, Page 675, 13, 675, https://doi.org/10.3390/RS13040675, 2021.

Gittings, J. A., Raitsos, D. E., Kheireddine, M., Racault, M. F., Claustre, H., and Hoteit, I.: Evaluating tropical phytoplankton phenology metrics using contemporary tools, Sci Rep, 9, https://doi.org/10.1038/s41598-018-37370-4, 2019.

Gittings, J. A., Raitsos, D. E., Brewin, R. J. W., and Hoteit, I.: Links between phenology of large phytoplankton and fisheries in the northern and central red sea, Remote Sens (Basel), 13, 1–18, https://doi.org/10.3390/rs13020231, 2021b.

Hauko et al. (2021) Phenology and Environmental Control of Phytoplankton Blooms in the Kong Håkon VII Hav in the Southern Ocean https://doi.org/10.3389/fmars.2021.623856

Jackson, Shubha Sathyendranathm, Frédéric Mélin et al (2017). An improved optical classification scheme for the Ocean Colour Essential Climate Variable and its applications https://doi.org/10.1016/j.rse.2017.03.036

Ji, R., Edwards, M., MacKas, D. L., Runge, J. A., and Thomas, A. C.: Marine plankton phenology and life history in a changing climate: Current research and future directions, J Plankton Res, 32, 1355–1368, https://doi.org/10.1093/plankt/fbq062, 2010.

Moutier, W.; Thomalla, S.J.; Bernard, S.; Wind, G.; Ryan-Keogh, T.J.; Smith, M.E. Evaluation of Chlorophyll-a and POC MODIS Aqua Products in the Southern Ocean. *Remote Sens.* 2019, *11*, 1793. https://doi.org/10.3390/rs11151793

Racault, M. F., Le Quéré, C., Buitenhuis, E., Sathyendranath, S., and Platt, T.: Phytoplankton phenology in the global ocean, Ecol Indic, 14, 152–163, https://doi.org/10.1016/j.ecolind.2011.07.010, 2012.

Racault, M. F., Sathyendranath, S., and Platt, T.: Impact of missing data on the estimation of ecological indicators from satellite ocean-colour time-series, Remote Sens Environ, 152, 15–28, https://doi.org/10.1016/J.RSE.2014.05.016, 2014.

Rubao Ji, Martin Edwards, David L. Mackas, Jeffrey A. Runge, Andrew C. Thomas, Marine plankton phenology and life history in a changing climate: current research and future directions, *Journal of Plankton Research*, Volume 32, Issue 10, October 2010, Pages 1355–1368, https://doi.org/10.1093/plankt/fbq062

Thomalla, Marie-Fanny Racault, Sebastiaan Swart, Pedro M. S. Monteiro, High-resolution view of the spring bloom initiation and net community production in the Subantarctic Southern Ocean using glider data, *ICES Journal of Marine Science*, Volume 72, Issue 6, July/August 2015, Pages 1999–2020, https://doi.org/10.1093/icesjms/fsv105

Thomalla, S. J., Nicholson, S. A., Ryan-Keogh, T. J., and Smith, M. E.: Widespread changes in Southern Ocean phytoplankton blooms linked to climate drivers, Nature Climate Change 2023 13:9, 13, 975–984, https://doi.org/10.1038/s41558-023-01768-4, 2023

Silva et al. 2021 Twenty-One Years of Phytoplankton Bloom Phenology in the Barents, Norwegian, and North Seas https://doi.org/10.3389/fmars.2021.746327

Tang, W., Llort, J., Weis, J. *et al.* Widespread phytoplankton blooms triggered by 2019–2020 Australian wildfires. *Nature* 597, 370–375 (2021). https://doi.org/10.1038/s41586-021-03805-8

Werther, M. Burggraaff, O. Dive Into the Unknown: Embracing Uncertainty to Advance Aquatic Remote Sensing. *J Remote Sens.* 2023;3:0070.DOI:10.34133/remotesensing.0070

---

## Author Comment (AC4)

We thank the reviewers for the time taken to review this work and for their insightful comments. Please see the responses below in blue.

**Reviewer report #1**

This study extracts indices about phytoplankton bloom from the ocean color dataset. As the authors pointed out, this approach has been used by several communities. Previous studies are well revisited; three different methods are adopted and cross-compared. Details about the method shown in the manuscript, that may help the other researchers who want to develop more advanced techniques. The indices are helpful to understand ecosystems and frequently adopted by many studies. Nonetheless, it was seldom provided as a dataset. In my opinion, this study and the datasets may be worth publishing in the Earth System Science Data once several concerns (especially those for Figure 4) are resolved. Below are my comments:

Figure 4 is wired for me. Figure 4b shows bloom duration longer than a year (larger than 400 days), that is not realistic and does not make sense for me. The SCR, that is a correlation (based on the definition stated in the manuscript), cannot exceed 1 (100%), nevertheless Figure 4c shows 1<SCR. The peak at SCR=1 (Figure 4c) is nonsense for me too. I presumed that some process in Figure 4 is not mentioned, or something is wrong here.

We thank the reviewer for highlighting these important concerns regarding Figure 4.

With regards to bloom duration longer than a year. To note, our bloom detection method does not constrain or force a bloom slice to be within a 12 month period, as is done in some other phenology studies e.g. Henson et al. 2018: *""For each calendar year we first identified the date of peak chlorophyll concentration and then concatenated the preceding and following 6 months."* So in their method a bloom can never be longer than 12 months. Instead, in our method, we recognise that there are areas where blooms can have multiple peaks. *"The 'bloom slice': The bloom slice, used to find the bloom initiation and termination dates, is identified for each pixel as the 6-month time span preceding and following from the maximum bloom peak (ii). Or in the case of multi-modal blooms, 6-months preceding the first and following the last peak respectively."* Typically blooms that have longer durations are found in oligotrophic gyres, characterised with low SCR (weak seasonality), highly variable and poorly defined blooms.

We have added the following text to clarify further, see section 3.1 in the manuscript:
*In these oligotrophic regions, where bloom amplitude is constrained by nutrients, the seasonality of phytoplankton blooms is not well-defined and characterised by high intraseasonal variability (Figure 2, Thomalla et al., 2011). Worth noting when applying our bloom detection method to these regions is that it does not constrain a bloom slice to be within a 12 month period, as is done in other phenology studies (e.g. Henson et al., 2018). Rather, by allowing for multiple peaks to be considered within a bloom, this approach may produce extended bloom durations that are beyond a year in regions with no discernable or strongly defined seasonal cycle.*

Furthermore, we have also added a new section: "***4 Limitations of the phenology algorithm and future developments"*** which highlights the limitations of such an approach, the detection methodology and future developments.

Regarding the SCR values that are 1. Over the entire dataset, all years, this only occurs in ~0.20% of the data. The reason for this is floating point errors when performing the regression between the climatological time series and the time series for each year. Below is a screenshot showing the maximum value of SCR, where the floating point errors manifest at the 16th decimal place.

```
print(f"{scr.max().values.item():.32f}")
```

1.00000000000000008881784197001 2523

To avoid any confusion we have removed any values where this floating point error occurs in the dataset.

Please see revised Figure 4. We have put the log-scale on the x-axis and unconstrained the y-axis for Figure 4 (a).

[Figure]

I would like to suggest using "satellite-driven" rather than "observational" in the entire manuscript including the title, because it can be easily misunderstood the data set using the in-situ observations. As the authors may know, satellite-driven measurements are occasionally not considered as observations due to the issues mentioned by the authors (gaps in the measurements and errors including bias in the algorithm). I think that "satellite-driven" is more clearly state the products in this study.

We have changed "observational" to "satellite-derived" throughout the text including the title of the manuscript. We chose satellite-derived over satellite-driven, we hope the reviewer agrees with this choice.

Minor comments:
L64: Typo? Not "Quay, 2017) Having" but "Quay, 2017). Having".
Noted and corrected. Thank you.

L158: Feel like that the abbreviation "SO" never been stated before. I presumed that it stands for Southern Ocean and suggest that do not use abbreviation.
Thank you for spotting this. We have removed the abbreviation and replaced with Southern Ocean

Figure 4a: Entire PDF (including the peak) for Bloom mean chl-a needs to be shown or, at least, stated. Log-scale axis or stating the information about the peak in caption may be helpful.
Updated figure 4. Please see above.

Figure A1: Is the time series from the in-situ observations provided by the stations or from the satellite measurement at the location of stations? This should be stated in either the caption of figure or the manuscript (maybe near L468).

We have updated the caption of Figure A1 to include: "Figure A1: Examples of phytoplankton bloom seasonal cycles of *satellite-derived chlorophyll-a from OC-CCI* and comparisons in phenological detection methods at….."

**Reviewer report #2**

This manuscript presents a satellite-derived chlorophyll-a dataset from the Ocean Colour Climate Change Initiative, providing phenological metrics at 4, 9, and 25 km spatial resolutions. The dataset is accessible and can be analysed easily using GIS/coding. The dataset is highly valuable for various research applications, including ecosystem monitoring, biodiversity assessments, and climate impact studies. The study is well conceived and has the potential to make a significant contribution to the field. Below, I provide some minor comments and suggestions for improvement: 1. A key concern is the lack of comparisons with prior phenology studies, particularly those utilizing in situ observations. While ship-based in situ measurements are indeed limited in spatial and temporal coverage, autonomous platforms such as BGC-Argo floats offer continuous data that could be utilized for such comparisons. Including a case study or analysis demonstrating agreement between satellite-derived and in situ-derived phenology metrics would enhance the dataset's credibility and highlight its utility. Additionally, discussing how the proposed dataset aligns with or diverges from earlier findings could provide valuable context. For reference, here are some relevant studies that might inform such comparisons (no need to cite; they are provided for your consideration):

For reference, here are some relevant studies that might inform such comparisons (no need to cite; they are provided for your consideration):

Demetriou, M., Raitsos, D. E., Kournopoulou, A., Mandalakis, M., Sfenthourakis, S., & Psarra, S. (2022). Phytoplankton Phenology in the Coastal Zone of Cyprus, Based on Remote Sensing and In Situ Observations. Remote Sensing, 14(1), 1–16. https://doi.org/https://doi.org/10.3390/rs14010012

Gittings, J. A., Raitsos, D. E., Kheireddine, M., Racault, M. F., Claustre, H., & Hoteit, I. (2019). Evaluating tropical phytoplankton phenology metrics using contemporary tools. Scientific Reports, 9(1), 1–9. https://doi.org/10.1038/s41598-018-37370-4

Kalloniati, K., Christou, E. D., Kournopoulou, A., Gittings, J. A., Theodorou, I., Zervoudaki, S., & Raitsos, D. E. (2023). Long-term warming and human-induced plankton shifts at a coastal Eastern Mediterranean site. Scientific Reports, 13(1). https://doi.org/10.1038/s41598-023-48254-7

Kournopoulou, A., Kikaki, K., Varkitzi, I., Psarra, S., Assimakopoulou, G., Karantzalos, K., & Raitsos, D. E. (2024). Atlas of phytoplankton phenology indices in selected Eastern Mediterranean marine ecosystems. Scientific Reports, 14(1), 9975. https://doi.org/10.1038/s41598-024-60792-2

Racault, M. F., Raitsos, D. E., Berumen, M. L., Brewin, R. J. W., Platt, T., Sathyendranath, S., & Hoteit, I. (2015). Phytoplankton phenology indices in coral reef ecosystems: Application to ocean-color observations in the Red Sea. Remote Sensing of Environment, 160, 222–234. https://doi.org/10.1016/j.rse.2015.01.019

We thank the reviewer for this great suggestion, and have added an additional paragraph which uses some of the examples provided above and others to make some regional comparisons with the data produced presented here and other phenology studies. See additional text (section 3.1):

*"A comparison of our satellite-derived phenology product with bloom indices derived from in situ data at a selection of regional case studies shows reasonable agreement. For example, in the Saronikos Gulf (Eastern Mediterranean), Kalloniati et al. (2023)  report a mean bloom initiation in early October (2005–2015), which*

*compares well with our mean bloom initiation over the same period of 24 September. Similarly, their mean bloom peak occurs in late February, closely matching our estimate of 24 of February. However, there are notable differences in bloom termination with their approach reporting a seasonal bloom that terminates in mid-April, compared to our estimate of ~100 days later on 13 July. This discrepancy likely arises because their method does not account for multiple bloom peaks, whereas our method is specifically designed to include the secondary peak observed in April as part of the seasonal bloom (see their Figure 3c). Another example from long-term mooring observations (1998-2022) in the Bering Sea shelf  (Nielsen et al. 2023)  reports the timing of the bloom maximum to range annually between the end of April to mid-June  (see their Figure 2), which compares well with our mean estimate over the same period of 25 of May (standard deviation of 57 days).  In a Red Sea comparison, although our satellite derived phenology data product was able to detect similar bloom initiation and max peak timing for the primary bloom in winter (as observed by Racault et al., 2015), it is not designed to provide indices fort bi-modal blooms and thus is unable to identify the secondary bloom in summer. Beyond these existing studies, we applied our phenological detection method (TS) to chlorophyll-a data from the HOT and BATS long-term monitoring sites (Figure 2A, Valente et al. 2022). At HOT (1998-2018)(Figure 2Aa), the in situ bloom initiation occurred on 25 July (±48 days) compared to the satellite-derived occurring on the 21 July (±42 days), in situ bloom max timing on 12th of December vs. 5th of December, and termination on 22 May (±32 days) vs. 6 June (±29 days) and duration in situ of 299 days vs durations of 303 days from satellite data. Similar agreement was seen in the BATS station (Figure 2Ab).*

In addition, we have used chlorophyll HPLC data from https://doi.pangaea.de/10.1594/PANGAEA.941318 to do some of our own in situ comparisons at the long-term monitoring sites HOT and BATS. See additional Figure A2 below:

[Figure]

*Figure A2: Comparison of five years of in situ chlorophyll-a measurements (Valente et al. 2022) with satellite-derived chlorophyll-a (OC-CCI), along with key phenological indices (solid and dashed vertical lines for satellite and mooring, respectively) at two sustained observing stations: (a) Hawaii Ocean Time-Series (HOT, 21° 20.6'N, 158° 16.4'W) and (b) Bermuda Atlantic Time-Series Study (BATS, 31° 50'N, 64° 10'W).*

2. Lines 498–501: Creating temporal composites is important for mitigating potential noise caused by interpolation errors in the OC-CCI dataset. While the product is highly valuable, it does exhibit some irregularities that can influence the calculation of phenology metrics, particularly at higher resolutions (e.g., 4 km), and therefore impact the derived phenology indicators described in this study. For example, when examining the spatial and temporal variability of phytoplankton growth period durations in the Indian Ocean (Figure 1), several regions (pixels) display durations ranging from approximately 100 to 600 days over the years. While such variability could be realistic in certain cases, it is important to recognize that, like any dataset, this product includes some irregularities that may affect its outputs.

[Figure]

We thank the reviewer for highlighting this important point. We have included the following additional section, which notes irregularities and limitations of the data product. As well as future developments for new releases of this product.

*4 Limitations of the phenology algorithm and future developments*

*The diversity of the phytoplankton seasonal cycles across the global ocean makes it challenging to generalise a single methodological approach that is capable of capturing all phenological metrics accurately. Our attempt to do so with this data product may lead to some irregularities, most notably when applied to regions with a poorly defined or unique seasonal cycle. For example, in ultra-oligotrophic regions where the bloom amplitude is particularly low and intraseasonal variability particularly high, our detection method prescribes long bloom durations that may exceed one year and can lead to overlapping bloom slices. Another example is regions with bi-modal blooms, where there is a well-defined summer and winter bloom in a given annual cycle. Although our phytoplankton phenology detection method is designed to allow for multiple peaks to occur within a bloom cycle; it has not been designed to cater for bimodal annual cycles, which would require the identification of separate summer and winter initiation and termination indices. In these instances our method may result in extended bloom durations. While these regions are relatively uncommon (e.g. Racault et al 2017, Figure 2c), they do exist, as is the case with the Red Sea (Racault et al. 2015). Future developments of this data product will endeavour to incorporate updates and improvements to the detection methods to better cope with these irregularities. We welcome users to reach out if other irregularities are identified within a specific area of*

*interest and to work with the authors to improve future versions of the product. All future changes to the product will be fully documented on Zenodo as new versions are released.*